# Artery formation in the intestinal wall and mesentery by intestine-derived Esm1+ endothelial cells

Esther Bovay [1], Kai Kruse [2], Emma C. Watson [1], Vishal Mohanakrishnan [1], Martin Stehling[3], Frank Berkenfeld[1], Mara E. Pitulescu [4], Mark L. Kahn[5] & Ralf H. Adams [1] ✉

Arterial blood transport into peripheral organs is indispensable for developmental growth, homeostasis and tissue repair. While it is appreciated that defective formation or compromised function of arteries is associated with a range of human diseases, the cellular and molecular mechanisms mediating arterial development remain little understood for most organs. Here, we show with genetic approaches that a small subpopulation of endothelial cells inside the intestinal villi of the embryonic mouse, characterized by the expression of endothelial cell-specific molecule 1 (Esm1/endocan), gives rise to arterial endothelium in the intestinal wall but also in the distant mesenteric vasculature. This involves cell migration but also substantial changes in morphology and gene expression. Immunohistochemistry and single cell RNA-sequencing confirm that intestinal Esm1+ cells have a distinct molecular profile and the capacity to undergo arterial differentiation. Genetic approaches establish that artery formation by the progeny of Esm1+ cells requires integrin β1 and signaling by the growth factor VEGF-C and its receptor VEGFR3. The sum of these findings demonstrates that Esm1+ cells inside the villus capillary network contribute to the formation of intestinal and mesenteric arteries during development.

Arteries are the conduit for blood transport into peripheral organs and thereby are crucial for the supply of essential nutrients and oxygen. Research in early mouse, zebrafish and avian embryos have provided a good understanding of the processes mediating the vasculogenic formation of the dorsal aorta and the cardinal vein, the two large axial vessels[1–3]. However, comparably little is known about the formation of the majority of arteries during the angiogenic expansion of the vasculature in the embryonic and postnatal organism. In the developing heart, it was shown that coronary artery growth involves the reprogramming of venous (sinus venosus-derived) endothelial cells (ECs) to an arterial fate[4]. However, endocardial cells, which form the innermost lining of the heart and share many features with arterial ECs, have been also shown to contribute to coronary artery development[5–8]. Genetic fate tracking has established that venous ECs, which exhibit comparably high rates of proliferation, give rise to arterial endothelium in retina and brain[9,10]. In the developing and regenerating zebrafish tail fin, dynamic live imaging has shown that vein-derived endothelial tip cells give rise to arterial ECs[11]. Strikingly, genetic alterations affecting guided EC migration impair normal arterial patterning, suggesting that defects in this process might be a cause of arteriovenous malformations[10,11].

[1]Max Planck Institute for Molecular Biomedicine, Department of Tissue Morphogenesis, Münster, Germany. [2]Max Planck Institute for Molecular Biomedicine, Bioinformatics Service Unit, Münster, Germany. [3]Max Planck Institute for Molecular Biomedicine, Flow Cytometry Unit, Münster, Germany. [4]Max Planck Institute for Molecular Biomedicine, Vascular Patterning Dynamics Group, Münster, Germany. [5]Cardiovascular Institute, Department of Medicine, Perelman School of Medicine, University of Pennsylvania, Philadelphia, PA, USA. ✉e-mail: ralf.adams@mpi-muenster.mpg.de

In the postnatal murine eye, tip cells at the distal end of endothelial sprouts, which express high levels of endothelial cell-specific molecule 1 (Esm1/endocan), are important for the angiogenic expansion of the developing retinal vasculature. Genetic fate tracking with *Esm1-CreERT2* transgenic mice has established that tip cell progeny contributes to the arterial but not the venous branch of the vasculature, which involves signaling interactions via the Notch pathway, the chemokine receptor CXCR4 and its ligand CXCL12, and interactions between ephrin-B2 and its receptor EphB4[11–13]. Arterial differentiation involves arrested proliferation and endothelial cell cycle state has been shown to generate a fate bias during arterial-venous specification[14,15]. Apart from molecular regulators, blood flow and fluid shear stress are important factors controlling cell cycle status and the migration of ECs against the direction of blood flow[14,16–19].

In the developing intestinal system of the mouse, arteriogenesis is initiated during gut rotation, at around embryonic stage E10.0, forming a first arterial connection between the intestinal vascular plexus with the dorsal aorta[20]. Like in other organs, the chemokine receptor CXCR4 and its ligand CXCL12 were shown to be essential in this process[21,22]. It also has been established that high signaling by vascular endothelial growth factor A (VEGF-A) and its receptor VEGFR2 upregulates the expression of Esm1 in the villus apex, which has relevance for nutrient uptake, blood vessel remodeling and normal arteriovenous patterning[19,23,24]. Despite of these insights, our understanding of the heterogeneity and functional specialization of ECs in the gastrointestinal system remains limited and the processes controlling artery formation in the intestine and adjacent mesentery are unknown. Our new findings reveal that a small subpopulation of ECs in the capillary network of the villus gives rise to arterial endothelium in the adjacent intestinal wall but also in the distant mesenteric vasculature. Genetic labeling, immunohistochemistry and single cell RNA-sequencing (scRNA-seq) show that these prearterial cells express Esm1 and other tip cell markers despite being located inside a patent capillary network. Mechanistically, *Esm1-CreERT2*-mediated inactivation of the *Itgb1* gene encoding integrin β1 in the mouse embryo results in the loss of arterial progenitors and a subsequent reduction in the diameter of large mesenteric arteries. Additionally, we show that the VEGF-C/VEGFR3 signaling pathway, which is a primary lymphatic growth factor and tyrosine kinase receptor[25,26], participates in the migration of Esm1+ cells into the mesenteric arterial network and expansion of the arterial tree. Taken together, these findings establish fundamental principles of developmental artery formation in the intestine and mesentery.

## Results

### Identification of arterial progenitors in the embryonic intestine

Stratification of the intestinal epithelium begins around embryonic day (E) 14.0[27–29]. However, the formation of the initial intestinal vascular network occurs earlier, beginning as early as E10.5[20]. We investigated whether the blood capillary network at these early stages already expresses Esm1. We stained E12.5 mesenteries and observed Esm1 protein expression within the capillary network surrounding the epithelium (Supplementary Fig. 1a). We further used *Esm1-CreERT2* transgenic mice in the *Rosa26-mTmG* Cre reporter background[30,31] to detect Esm1+ cells and their progeny. After tamoxifen treatment initiated at E10.5, recombined GFP+ ECs appear in the SOX17+ intestinal and mesenteric vascular network at E13.5 (Supplementary Fig. 1b).

We further investigated whether the early mesenteric vasculature already shows segregation of arteries and veins. To analyze known arterial and venous markers, respectively, we used *Aplnr-CreERT2* and *Bmx-CreERT2* transgenic mice in combination with the *Rosa26-mTmG* Cre reporter[30,32,33]. After daily tamoxifen administration from E10.5 onward, *Bmx-CreERT2*-controlled GFP expression was observed in the Endomucin-negative arterial network of E13.5 mesenteric tissues including the cranial mesenteric artery (CMA) (Supplementary Fig. 1c).

Conversely, 24-h *Aplnr-CreERT2* induction with 4-hydroxytamoxifen (4-OHT) resulted in widespread GFP expression in E12.5 mesenteric and intestinal vessels, whereas SOX17+ vessels connected to the CMA remained GFP<sup>low</sup> (Supplementary Fig. 1d). These findings indicate that Esm1+ cell progeny contributes to arteries during an early stage of intestinal and mesenteric development. Furthermore, it is evident that acute *Aplnr-CreERT2*-mediated recombination in the E11.5 embryo is prominent in capillaries but spares BMX+ SOX17+ mesenteric arteries.

We next examined Esm1 expression at later stages. Whole-mount staining of embryonic tissues at E18.0 revealed strong Esm1 protein expression in the intestinal villi and mesenteric capillary sprouting cells (Fig. 1a). To assess the long-term contribution of Esm1+ cells, we treated *Esm1-CreERT2* transgenic mice in the *Rosa26-mTmG* Cre reporter background from E10.5 to E15.5 (Fig. 1b). GFP+ ECs were abundant in large mesenteric arteries and smaller arterial branches near the intestinal wall at E16.5 (Fig. 1b) but absent from veins or lymphatic vessels. To gain a better insight into the spatial distribution and dynamic behavior of Esm1+ cells, we conducted multiple genetic fate tracking experiments (Fig. 1c). Short-term (24 h) induction with 4-hydroxytamoxifen (4-OHT) leads to robust GFP signal in the embryonic intestine, while the mesentery exhibits minimal presence of GFP+ ECs, except for mesenteric capillaries and mesenteric sprouting ECs (Fig. 1d–f, Supplementary Fig. 1e). Analysis of intestinal villi at E18.5 revealed that GFP signal is predominantly localized to the upper SOX17+ portion of the villus vascular network, reflecting the previously reported heightened VEGF signaling at the apex of the villus[24] (Fig. 1g, h). Analysis of embryos at the same stage but after 3 and 5 days post-induction shows lower GFP signal within the intestine but an increased presence of GFP+ cells within large mesenteric arteries (Fig. 1d–f). While the extent of arterial labeling is limited in these short-term genetic fate tracking experiments, the results indicate that *Esm1-CreERT2*-labeled cells are incorporated into mesenteric arteries but not into nearby veins or lymphatic vessels (Fig. 1d–f).

Careful analysis of GFP+ cells over time indicates morphological changes during the transition from the intestine into the mesentery. Whereas GFP+ ECs in the intestine display irregular shapes with numerous protrusions, cells in the mesentery exhibit the typical slender and elongated morphology of arterial ECs (Fig. 1e, g). With increasing time between 4-OHT administration and analysis, more elongated arterial GFP+ cells are observed, and these cells are found deeper inside the mesenteric arterial tree (Fig. 1d, e). Further analysis of intestines and quantitation confirm the progressive decline of GFP+ Esm1+ EC progeny in the distal villus area along with an increasing contribution to the SOX17+ villus capillaries, submucosal arteries and mesenteric arteries over time (Fig. 1d–i).

As these short-term cell tracing experiments label a limited number of Esm1+ cell progeny within the arterial network, we validated that GFP+ ECs do not enter apoptosis (Supplementary Fig. 2a, b). Few cells labeled for activated caspase-3 are present in the E14.5 intestine, but the marker of apoptosis is not found inside the vasculature or in GFP+ cells (Supplementary Fig. 2a, b). Short-time (10 h) lineage tracing comparing GFP+ cells and endogenous Esm1 protein expression shows that the latter is more widespread, indicating incomplete *Esm1-CreERT2* expression or 4-OHT-induced activation (Supplementary Fig. 2c–e). Taken together, the sum of our genetic tracking data indicates that Esm1+ cells continue to be induced throughout development of the embryonic intestinal vasculature from midgestation to birth. A single pulse of tamoxifen/4-OHT will lead to *Esm1-CreERT2*-mediated recombination only in a fraction of these cells.

### Venous origin of intestinal arterial progenitors in the embryonic intestine

In the mouse retina and embryonic heart, venous cells actively divide and give rise to all other EC subpopulations including tip cells and committed arterial ECs that exhibit cell cycle arrest[4,14,15,34,35]. Following

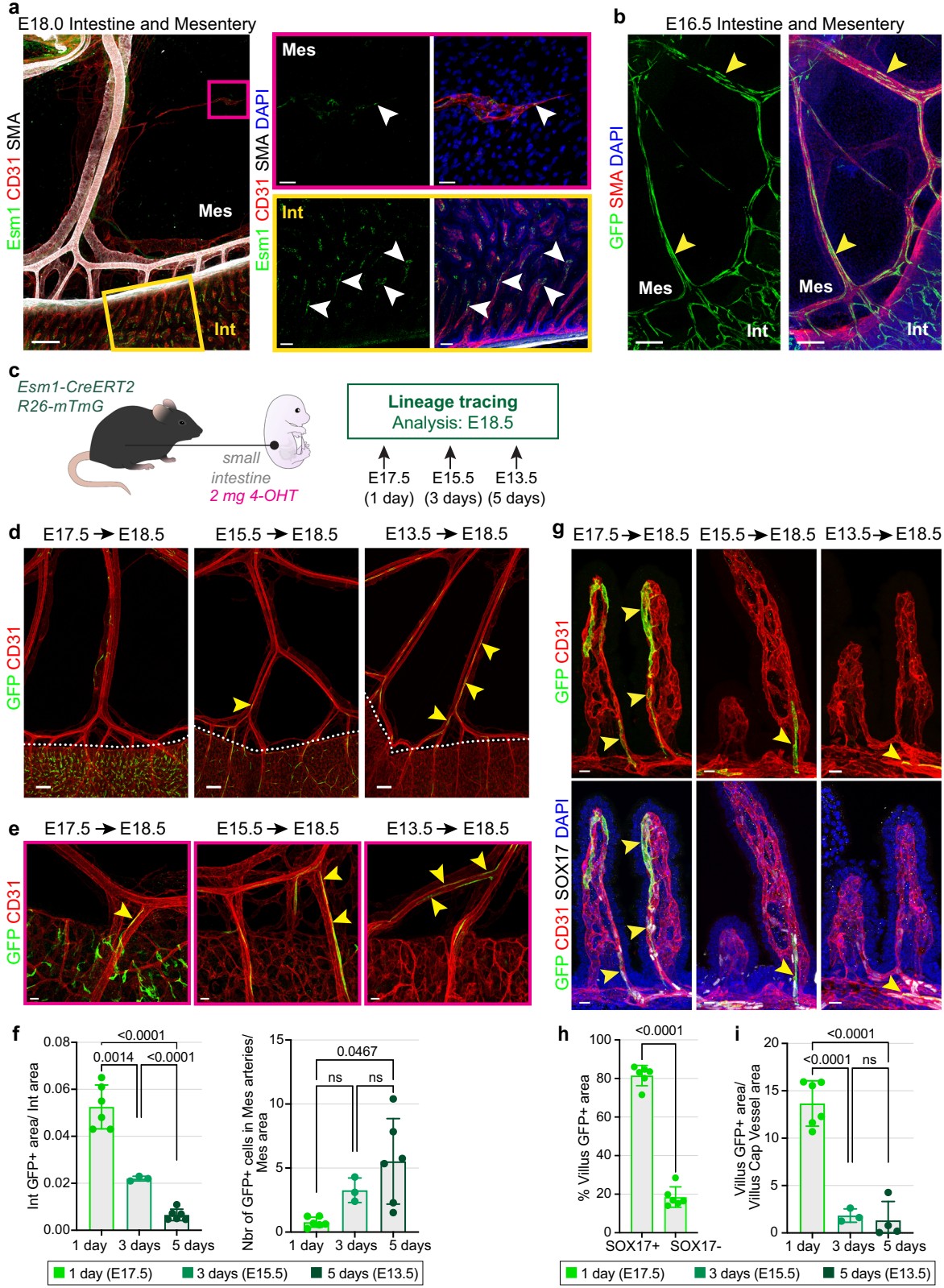

this logic, we hypothesized that venous ECs contribute to the Esm1+ cell population, which subsequently gives rise to the Bmx+ arterial network. Conversely, fully differentiated Bmx+ cells are unlikely to revert to Esm1+ cells or generate new Esm1-derived arterial endothelial cells. To test this hypothesis, we used the tamoxifen-inducible *Bmx-CreERT2* mouse line[33] in combination with the *R26-mTmG* reporter. In lineage tracing experiments with the common end point at E18.5, GFP signal labels large mesenteric arteries and intestinal submucosal SOX17+ arteries (Supplementary Fig. 3a–d). Over longer time periods, namely after 4-OHT induction at E13.5 or E15.5, GFP-negative ECs replace the *Bmx-CreERT2*-labeled GFP+ cells in submucosal and smaller (distal) mesenteric arteries so that recombined ECs reside only in the larger caliber arteries (Supplementary Fig. 3b). These results strongly support the notion that the continuous migration of Bmx-negative

**Fig. 1 | Generation of intestinal arterial progenitors during embryonic development. a** Whole-mount of E18.0 mesentery and intestine shows that Esm1 is expressed in mesenteric sprouting cells (pink box) and intestine villus capillaries (yellow box). Esm1 (green), CD31 (red), SMA (white) and DAPI (blue). Arrowheads point at Esm1+ ECs. Mes: Mesentery, Int: intestine. $n = 4$. Scale bars, 200, 50 and 30 μm. **b** Esm1+ cell-derived GFP+ (green, arrowheads) ECs contribute to large mesenteric arteries (SMA, red) at E16.5. DAPI (blue). Mesentery (Mes) and intestine (Int) are indicated. $n = 3$. Scale bar, 150 μm. **c** Experimental design of Esm1+ cell lineage tracing. **d** Whole-mount of E18.5 mesentery and intestine at 1 day, 3 days and 5 days of lineage tracing. GFP (green) and CD31 (red). Arrowheads mark GFP+ arterial ECs. 1 day $n = 6$; 3 days $n = 3$; 5 days $n = 6$. Scale bar, 200 μm. **e** High-magnification images of E18.5 mesentery and intestine after lineage tracing; GFP (green) and CD31 (red). Arrowheads, GFP+ cells in mesenteric arteries. Scale bar, 30 μm. **f** Quantification of GFP+ area in the intestinal tissue (μm$^2$, normalized to tissue area) and number of arterial GFP+ ECs in mesenteric tissue (normalized to tissue area). P values, Brown-Forsythe and Welch ANOVA with Dunnett post-hoc test; Error bars, Mean ± SD. **g** Whole-mount of E18.5 intestinal villi at 1 day, 3 days and 5 days of lineage tracing. GFP (green), CD31 (red), SOX17 (white) and DAPI (blue). Arrowheads mark GFP+ cells descending into the SOX17+ intestinal arterioles. Scale bar, 15 μm. **h** Proportion of GFP+ area in SOX17+ or SOX17− villus capillaries (μm$^2$). 1 day $n = 6$. P values, 2-tailed unpaired Student's $t$ test; Error bars, Mean ± SD. (**i**) GFP+ area in the villus capillary network (μm$^2$, norm by vessel area). 1 day $n = 6$; 3 days $n = 3$; 5 days $n = 4$. P values, 1-way ANOVA with Tukey post-hoc test; Error bars, Mean ± SD.

arterial progenitors from the intestine contributes to the expansion of embryonic mesenteric arteries.

To investigate the origin of Esm1+ cells and the cells generating upstream arteries, we performed lineage-tracing of Aplnr+ ECs (Supplementary Fig. 4). To validate the expression of *Aplnr* as a marker of the venous domain, we used the *Aplnr-CreERT2* line in combination with the *Rosa26-mTmG* Cre reporter[30,32]. At 24 h after 4-OHT administration, *Aplnr-CreERT2*-controlled GFP expression is detected in the mesenteric veins and capillaries but is excluded from SOX17+ arteries and lymphatic vessels (Supplementary Fig. 4a−d). Inside the intestine, GFP signal is found throughout the majority of the villus capillary network but is excluded from the SOX17+ branch connecting to the submucosal arteries (Supplementary Fig. 4k). At 4−6 days after 4-OHT treatment, Aplnr+ EC-derived cells contribute strongly to the villus capillary network, the SOX17+ submucosal arteries, and large upstream mesenteric arteries (Supplementary Fig. 4e−k). Notably, lineage tracing from E13.5 leads to labeling of large veins, capillaries, and arteries closer to the intestine, while the majority of lymphatic vessels lacks GFP expression. However, when 4-OHT treatment is initiated at E11.5, nearly all vessels, including lymphatic vessels, are GFP+ (Supplementary Fig. 4e−k). These findings are consistent with previous research indicating that Etv2+ mesenchymal cell-derived angioblasts and venous ECs give rise to mesenteric lymphatic vessels in the mouse[36–38]. The results also support that Esm1+ cells and arterial ECs are derived from Aplnr+ ECs.

To further validate our intestinal and mesenteric artery model, we examined Esm1+, Bmx+, and Aplnr+ cell-derived progeny, respectively, following 4-OHT induction at E13.5. After 24 h, *Esm1-CreERT2*-labeled cells are rarely present in mesenteric tissue and localize mainly to the intestine (Supplementary Fig. 5a). By day 5, however, GFP+ Esm1+ cell progeny is notably enriched in the mesenteric arteries (Supplementary Fig. 5a, b). Using a similar strategy, *Bmx-CreERT2*-controlled GFP expression at E14.5 decorates mesenteric arteries at 24 h after 4-OHT administration (Supplementary Fig. 5c). By day 5, at E18.5, GFP-negative arterial domains can be seen in the submucosa and mesentery near the intestine, consistent with the incorporation of unlabeled pre-arterial ECs (Supplementary Fig. 5c, d). *Aplnr-CreERT2* short-term tracking leads to labeling of mesenteric capillaries and veins at E14.5 (Supplementary Fig. 5e), whereas GFP+ progeny is present in submucosal and mesenteric arteries adjacent to the intestine at E18.0 (Supplementary Fig. 5e, f). Together, these results support a model in which venous-derived (Aplnr+) ECs give rise to Esm1+ cells, which, in turn, differentiate into Bmx+ arterial ECs and thereby progressively contribute to the mesenteric arterial network (Supplementary Fig. 5g).

### Intestinal arterial progenitors contribute to the villus capillary expansion after birth

To investigate whether progeny from Esm1+ ECs continues to contribute to arteries after birth, lineage tracing experiments were initiated from postnatal day (P) 1 (Supplementary Fig. 6a). Analysis at P8 reveals an abundance of GFP+ ECs within the intestine, whereas the

mesentery lacks GFP-expressing cells in large arteries. GFP+ cells are located in the expanding capillary network surrounding mesenteric arteries and veins but lack a direct connection to these large vessels (Supplementary Fig. 6b). Further examination of intestines at P8 and P21 confirms the presence of GFP+ cells in the SOX17+ region of the villus capillary network (Supplementary Fig. 6c, d). These results indicate the continued presence of Esm1+ cells in the intestinal vasculature after birth, but these cells no longer contribute to large mesenteric arteries.

Previous work has shown that high VEGF signaling at the villus apex, indicated by elevated expression of the VEGF-responsive genes *Flt4* (encoding VEGFR3) and *Esm1*, facilitates the reorganization of EC junctions to enhance nutrient uptake in the adult mouse intestine[24]. Consistent with these findings, whole-mount analysis of the adult mouse intestine shows Esm1 immunostaining in the peri-arterial domain of the villus capillary network, which is characterized by pronounced anti-Caveolin 1 signal and absent Endomucin expression (Supplementary Fig. 7a, b). Lineage tracing experiments conducted over 24 h and 2 weeks in 20 to 24-week-old male mice reveal only minimal presence of GFP+ ECs in the retina of these animals (Supplementary Fig. 7c−e). In contrast, robust GFP signal is seen inside the capillary network of intestinal villi, with a significant enrichment in Endomucin-negative areas (Supplementary Fig. 7f−h). Altogether, these findings show that the expression of Esm1 in the intestinal vascular network commences during embryogenesis and persists into postnatal and adult stages. However, the contribution of Esm1+ EC-derived arterial progenitors to the formation of mesenteric arteries is confined to embryonic development.

### Fate and properties of intestinal arterial progenitors at single cell resolution

To validate the developmental fate of intestinal Esm1+ ECs and gain deeper insight into their molecular properties, we isolated cells from E18.0 embryos 24 h after genetic lineage tracing with *Esm1-CreERT2* (Fig. 2a, Supplementary Fig. 8). We first isolated all cells separately from the intestine and mesentery, followed by single-cell RNA sequencing (scRNA-seq) using the BD Rhapsody system. To augment our population of blood vessel ECs, we conducted an additional 24 h lineage tracing experiment with ECs enriched in GFP+ cells by fluorescence-activated cell sorting (FACS) (Supplementary Fig. 8a, b). In total, we acquired 34518 cells, among which 7122 were blood vessel ECs (Supplementary Fig. 8b). These datasets were combined and visualized as Uniform Manifold Approximation and Projection (UMAP) plots (Fig. 2b, c).

Based on established markers from the literature, we identified most of the known intestinal and mesenteric cell types, including ECs from blood and lymphatic vessels, epithelial, mesothelial, immune, neural and various mesenchymal cells (Fig. 2d, e). We then focused on blood vessel ECs, which segregate into 8 subgroups (after exclusion of contaminating red blood cells) (Fig. 2c, e). ECs expressing higher levels of venous markers, such as *Nrp2, Madcam1, Aplnr* and *Nr2f2* (Coup-TFII), can be seen on the left side of the UMAP plot, subdivided into

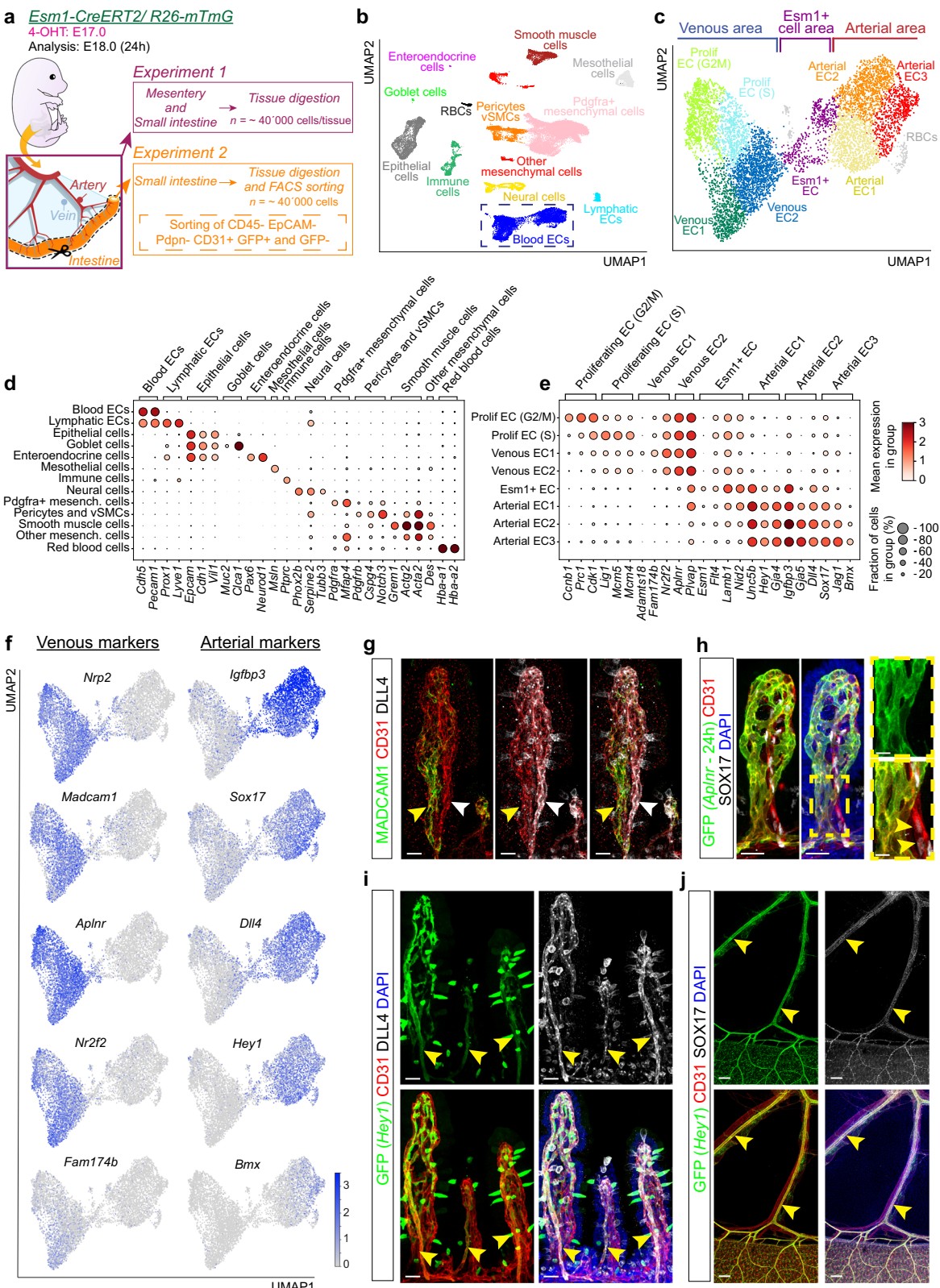

proliferative (G2/M and S phase) and two non-proliferative groups (Venous EC1 and Venous EC2). ECs with higher levels of arterial markers, namely *Unc5b*, *Hey1*, *Gja4*, *Igfbp3*, *Gja5*, *Dll4* and *Sox17*, cluster on the right (Arterial EC1, Arterial EC2, and Arterial EC3). Venous EC1 and Arterial EC3 cells are enriched in ECs from larger veins and arteries from the intestinal submucosa and mesentery, expressing markers such as *Adamts18*, *Fam174b* and *Bmx*, respectively (Fig. 2e, f,

Supplementary Fig. 8b, c). Moreover, these two populations lack capillary markers, such as *Pcdh12*, *Gpihbp1* and *Prdm1* (Supplementary Fig. 8c), which indicates that they are most mature with respect to arteriovenous specification. Interestingly, an intermediate subgroup of ECs is located in the center between the areas expressing venous or arterial markers. This population shows high levels of markers characteristic for endothelial tip and capillary cells (*Esm1*, *Flt4*, *Lamb1*, *Nid2*,

**Fig. 2 | Endothelial cell zonation in the embryonic intestine and mesentery.**
**a** Experimental design of the scRNA-seq experiments. Unsorted total intestinal cells, unsorted total mesenteric cells and FACS-sorted intestinal blood ECs were individually sequenced and the resulting datasets were integrated (unsorted samples, $n = 3$ pooled embryos; FACS-sorted sample, $n = 13$ pooled embryos). UMAP plot of the 3 combined datasets with all identified intestinal and mesenteric cell types (**b**) or blood vessel ECs only (**c**). **d**, **e** Dot plots of known marker genes for each cell population. Color code indicates average expression level in each group, and the dot size indicates the percentage of cells in each group expressing the given gene. **f** UMAP plots showing the expression of venous and arterial markers in the blood vessel EC populations. **g** Whole-mount of E18.5 intestine showing that MADCAM1 (green, yellow arrowheads) and DLL4 (white, white arrowheads) immunostaining marks the venous and arterial side of the villus capillary network (CD31, red), respectively. $n = 4$. Scale bar, 20 μm. **h** Whole-mount of E17.0 villus at 24 h of *Aplnr* lineage tracing; GFP (green), CD31 (red), SOX17 (white) and DAPI (blue). Arrowheads mark SOX17+ GFP- ECs. $n = 4$. Scale bars, 30 and 10 μm. **i** Whole-mount of E18.5 *Hey1-eGFP* intestines. GFP (green), CD31 (red), DLL4 (white) and DAPI (blue). Arrowheads, GFP+ DLL4+ vessels. $n = 6$. Scale bar, 30 μm. **j** Whole-mount of E18.5 *Hey1-eGFP* embryonic mesenteries. GFP (green), CD31 (red), SOX17 (white) and DAPI (blue). Arrowheads indicate GFP+ SOX17+ arteries. $n = 6$. Scale bar, 200 μm.

*Pcdh12*, *Gpihbp1* and *Prdm1*) and represents Esm1+ ECs (Fig. 2c, e; Supplementary Fig. 8b, c).

We further corroborated the distribution of venous and arterial markers at the protein level by immunohistochemistry (Fig. 2g–j). Staining of E17.0-E19.0 embryonic intestines confirms the expression of MADCAM1 and DLL4 by the peri-venous and peri-arterial side of the villus capillary network, respectively (Fig. 2g). As already shown (Supplementary Fig. 4k), 24 h lineage tracing for the venous marker Aplnr yields no GFP signal in the SOX17+ villus capillary branch (Fig. 2h). On the other hand, analysis of a *Hey1-eGFP* transgenic reporter mouse line shows more prominent GFP signal in the peri-arterial, SOX17+ side of the villus capillary network and in large mesenteric arteries, reflecting the known role of Notch signaling in arterial specification (Fig. 2i, j). Consistent with previous reports, *Hey1-eGFP* expression is also detected in some epithelial cells and DLL4+ ECs[39–42]. Taken together, these findings validate our scRNA-seq data and confirm arteriovenous zonation within the developing villus vasculature, which gradually extends into the large veins and arteries of the mesenteric vascular network.

## Transient cell cycle arrest and arterial differentiation of vein-derived Esm1+ cells

Consistent with this and previous studies, the venous EC subclusters in our scRNA-seq analysis predominantly harbor proliferative cells in the G2/M or S phase (Fig. 3a). To exclude clustering effects due to dominant cell cycle expression patterns, we regressed out the expression of known G2/M and S phase genes, re-clustered the cells, and annotated the new clusters to match the original EC type annotation as closely as possible (Supplementary Fig. 9a). We then again predicted the proportion of cycling cells in each EC population, by using Tricycle (version 1.16.0)[43] (Supplementary Fig. 9b, c), which confirmed the high proliferation of venous ECs.

To validate these observations, we conducted a Partition-based Graph Abstraction (PAGA) Trajectory analysis, facilitating diffusion pseudotime calculation[44] (Supplementary Fig. 9d). For this analysis, the two proliferative subgroups, i.e. Prolif EC (G2/M phase) and Prolif EC (S phase), were excluded. Pseudotime results are consistent with a trajectory along the vein-to-artery axis, extending from the venous side through the Esm1+ population into the arterial domain (Supplementary Fig. 9e). As expected, the Venous EC1 and Venous EC2 subgroups show low expression of arterial markers but high levels of cell cycle genes relative to other EC subpopulations (Fig. 3b, c). Conversely, Esm1+ ECs, Arterial EC1 and Arterial EC2 exhibit higher expression of arterial markers but low transcript levels for cell cycle genes, confirming previous observations[4,14,15,34,35,45]. The Arterial EC3 subgroup, which is enriched in the mesenteric sample and therefore represents ECs from larger arteries, surprisingly shows higher levels of cell cycle genes compared to other arterial subpopulations (Fig. 3b, c, Supplementary Fig. 9b). We validated these findings with EdU staining to assess EC proliferation in the intestine and mesentery. The majority of EdU+ ERG+ ECs in villi lack SOX17 expression, reflecting the expected low proliferation of arterial progenitors undergoing differentiation (Fig. 3d–f). In large mesenteric arteries, however, over 10% of ERG+ elongated

arterial nuclei are EdU+. Taken together, these results indicate that Esm1+ cells are derived from venous ECs in the developing intestine. Esm1+ cell progeny shows the expected downregulation of cell cycle genes during arterial specification. In contrast, a fraction of fully differentiated mesenteric arterial ECs re-express cell cycle genes, suggesting that local proliferation of these cells contributes to the size increase of larger arteries during development.

## Comparison of Esm1+ ECs in the embryonic villus and postnatal retina

As previous work has shown that Esm1+ ECs in the postnatal retina contribute to the expanding arterial network[12,13], we compared the molecular properties of these cells to the Esm1+ population from embryonic intestine. To this end, we reanalyzed a previously published scRNA-seq dataset of wildtype P6 retinal endothelial tip cells[46] and compared the Esm1+ population to the Esm1+ cluster in our study using pseudobulk differential expression analysis (DEA) (Supplementary Fig. 10a). While Esm1+ cells from the retina and embryonic intestine share expression of signature genes such as *Esm1*, *Flt4* (encoding the receptor tyrosine kinase VEGFR3) or *Lama4* (laminin α4), the two populations show substantial differences in gene expression. One example is the *Fabp4* gene and its product fatty acid binding protein 4, which are highly expressed by Esm1+ ECs in the distal villus but absent in retinal tip cells[47,48] (Supplementary Fig. 10b–d). Whole-mount immunostaining shows substantial morphological differences between Esm1+ cells in the retina and intestine (Supplementary Fig. 10c–f). While intestinal Esm1+ cells are an integral part of the villus capillary network, Esm1+ cells in the retina are sprouting tip cells that extend filopodia into the avascular tissue (Supplementary Fig. 10c–f). Furthermore, Caveolin-1 (encoded by the *Cav1* gene), is differentially expressed between the intestinal and retinal Esm1+ populations by scRNA-seq (Supplementary Fig. 10b). Immunostaining confirms low expression of Caveolin-1 in sprouting retinal tip cells, whereas Esm1+ cells in the capillary network of intestinal villi are prominently labeled (Supplementary Fig. 10e, f).

Whole-mount immunostaining of the embryonic intestine at E18.0/E18.5 confirms that endogenous Esm1 protein expression is enriched in the upper portion of the villus, in contrast to ubiquitous CD31 expression by all ECs (Fig. 4a, b). *Esm1-CreERT2*-mediated lineage tracing for 24 h in combination with immunostaining shows that *Esm1* expression precedes the upregulation of SOX17, thus affirming the role of Esm1+ ECs as arterial progenitors (Fig. 4c). Analysis of genetic lineage tracing over 24 h by scRNA-seq provides further proof of the contribution of *Esm1-CreERT2*-labeled (GFP+) progeny from the central Esm1+ population to the arterial branch of the intestinal vasculature (Fig. 4d). We then performed immunostaining for the markers laminin α4, collagen IV and VEGFR3 (Fig. 4e–g). Indeed, all three gene products were found to decorate the capillaries in the distal villus, confirming that the microenvironment at the apex of the embryonic villus facilitates the emergence of Esm1+ ECs with a role in artery formation (Supplementary Fig. 10g).

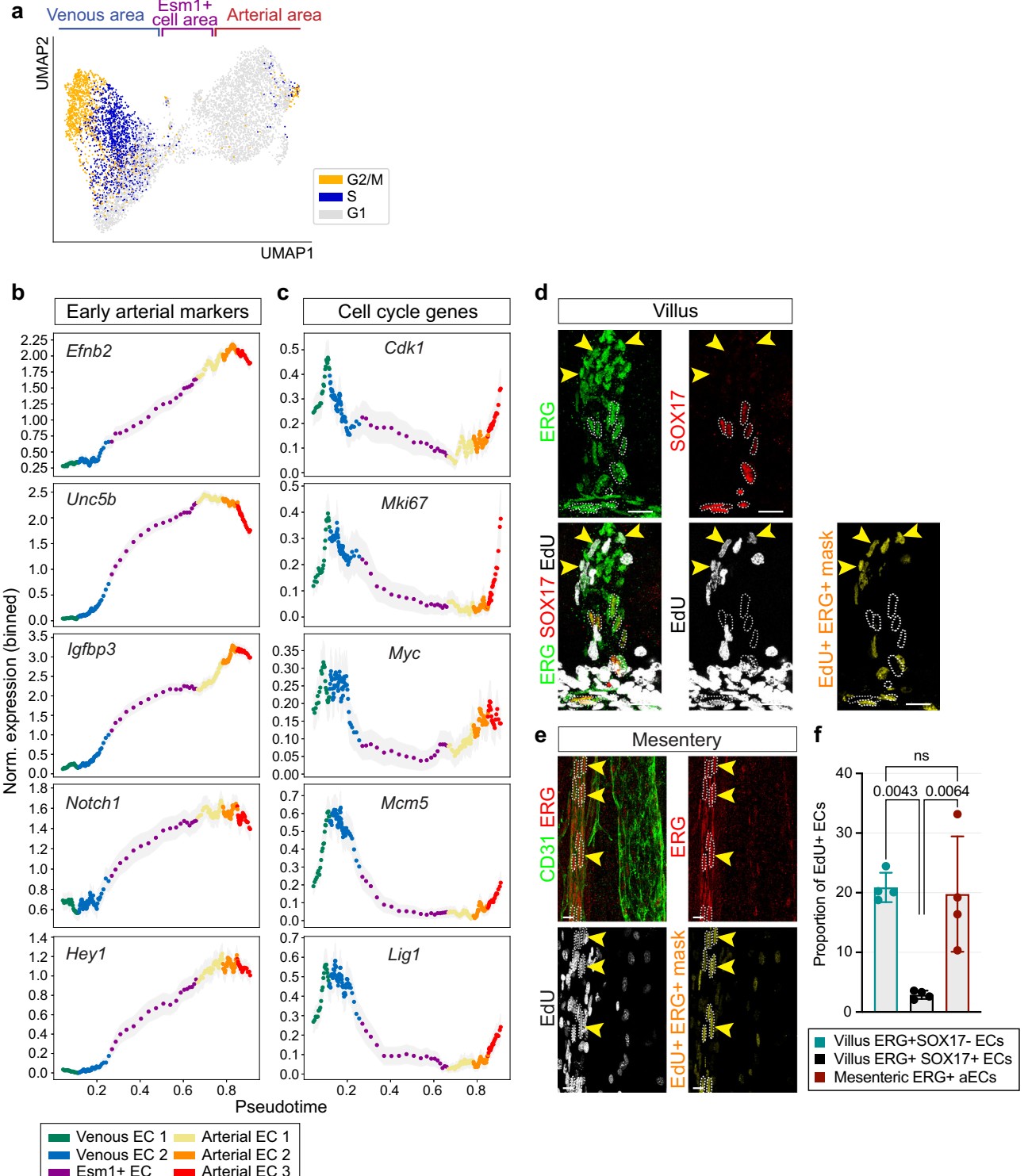

**Fig. 3 | EC proliferation during and after arterial specification. a** UMAP plot of ECs determined to be in G2/M, S and G1 phase. **b** Increase of early arterial gene expression along the vein-to-artery pseudotime axis. **c** Downregulation of cell cycle genes in the Esm1⁺ EC, Arterial EC1 and Arterial EC2 subgroups. **d** SOX17⁺ villus capillary ECs have lower proliferation compared to SOX17⁻ ECs. Whole-mount of E18.5 intestine with ERG (green), SOX17 (red) and EdU (white) staining. A mask was applied to identify EdU⁺ ERG⁺ cells (yellow). Arrowheads indicate EdU⁺ ERG⁺ cells.

$n = 4$. Scale bar, 20 μm. **e** Proliferation of large mesenteric arterial ERG⁺ ECs. Whole-mount of E18.5 mesentery with CD31 (green), ERG (red) and EdU (white) staining. A mask was applied to identify EdU⁺ ERG⁺ cells (yellow). Arrowheads mark EdU⁺ ERG⁺ cells. $n = 4$. Scale bar, 15 μm. **f** Proportion of EdU⁺ ERG⁺ SOX17⁺ and EdU⁺ ERG⁺ SOX17⁻ ECs in the intestine and EdU⁺ ERG⁺ ECs in the mesenteric artery. $n = 4$ for each tissue. P values, 1-way ANOVA with Tukey post-hoc test; Error bars, Mean ± SD.

## Integrin β1 in Esm1⁺ ECs controls cell shape and mesenteric artery diameter

Gene Ontology (GO) enrichment analysis of genes exhibiting a similar expression pattern as *Esm1* revealed a significant enrichment of genes involved in the biological processes "angiogenesis" and "cell migration". Likewise, "Focal adhesion" and "Actin filament" were among the top hits for cellular components (Supplementary Fig. 11a). To address a potential role of cell migration and cell-matrix interactions, we used

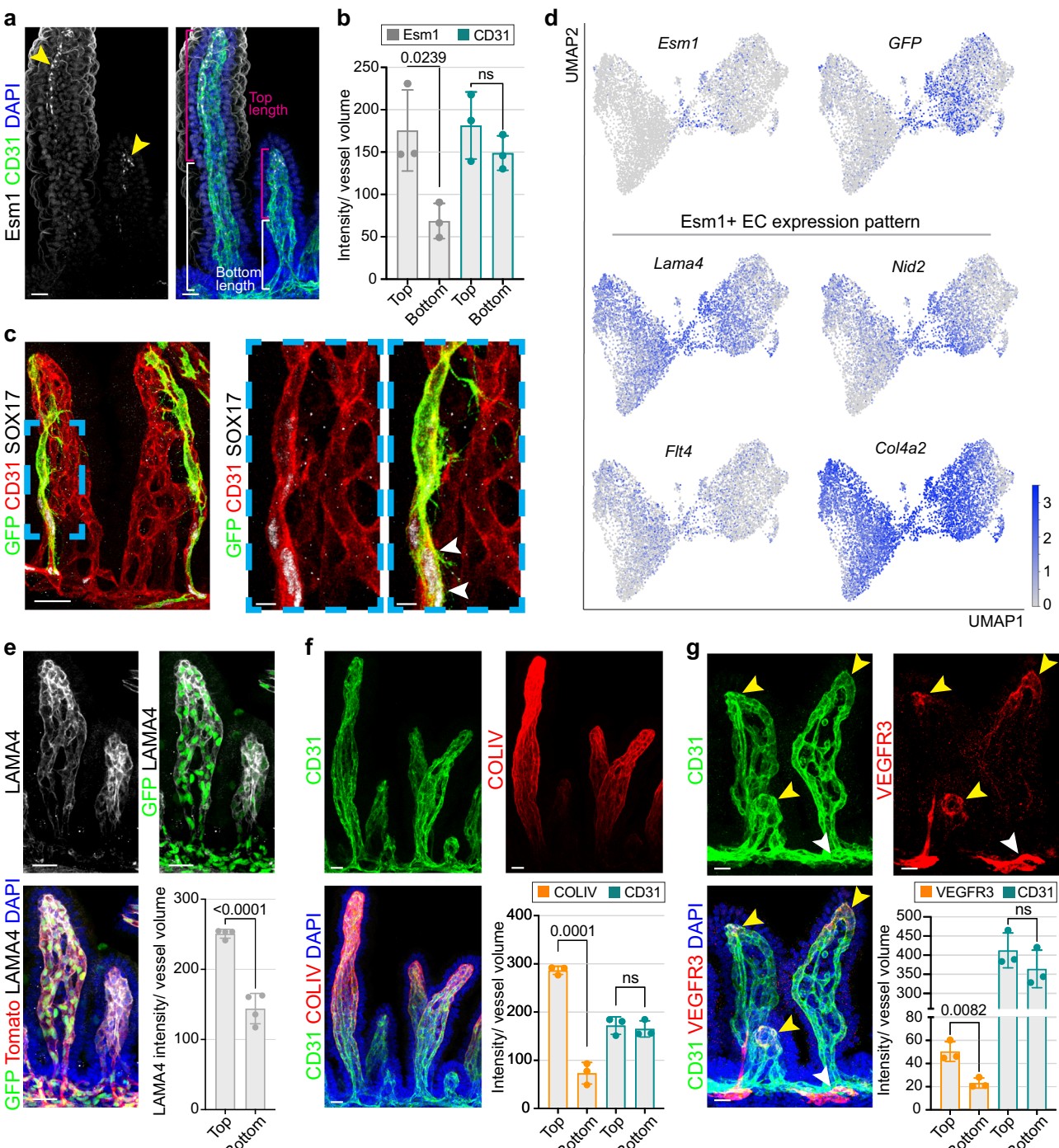

**Fig. 4 | Intestinal arterial progenitor cells express tip cell-like markers. a** Whole-mount of E18.0 intestine shows expression of Esm1 (arrowheads) in the apical part of the villus capillary network. Esm1 (white), CD31 (green) and DAPI (blue). $n = 3$. Scale bar, 20 μm. **b** Quantification of Esm1 and CD31 intensity (A.U.) inside villi normalized to blood vessel volume (μm³). P values, 2-tailed unpaired Student's $t$ test; Error bars, Mean ± SD. **c** Genetic lineage tracing with *Esm1-CreERT2* (24 h) shows recombined GFP⁺ cells in the arterial side of the villus vasculature in the E18.5 intestine. GFP (green), CD31 (red) and SOX17 (white). White arrowheads mark GFP⁺ SOX17⁺ ECs. $n = 6$. Scale bars, 30 and 7 μm. **d** UMAP plots showing the expression levels of *Esm1, GFP, Lama4, Col4a2, Flt4* and *Nid2*. *Esm1-CreERT2*-induced *GFP* expression (24 h after 4-OHT treatment) is highest in the Esm1⁺ population and the arterial cell area. **e** Whole-mount *Cdh5-mTnG* intestines at E18.0 (GFP marks EC

nuclei in green and mTomato EC surface in red) shows Laminin α4 (LAMA4) immunostaining (white) in the upper part of the villus vasculature. Nuclei, DAPI (blue). $n = 4$. Scale bar, 30 μm. Graph shows quantification of LAMA4 intensity (A.U.) in villi normalized to vascular volume (μm³). P values, 2-tailed unpaired Student's $t$ test; Error bars, Mean ± SD. **f, g** Increased Collagen type IV (COLIV) and VEGFR3 expression in the upper part of the villus vasculature in the E18.0-E18.5 intestine. CD31 (green), COLIV or VEGFR3 (red) and DAPI (blue). Yellow arrowheads mark VEGFR3^high blood vessels, white arrowhead indicates VEGFR3^high lymphatic vessel. $n = 3$. Scale bars, 20 μm. Graphs show quantification of COLIV and VEGFR3 intensity (A.U.) in villi normalized to vascular volume (μm³). P values, 2-tailed unpaired Student's $t$ test; Error bars, Mean ± SD.

the *Esm1-CreERT2* line to inactivate the *Itgb1* gene, which encodes the integrin β1 subunit (Fig. 5a). Integrins are heterodimeric transmembrane receptors composed of an α and β subunit[49–52]. Integrin β1 plays a crucial role in the development and maintenance of the blood vascular network. Pan-endothelial deletion of *Itgb1* was shown to result in vascular leakage and the disruption of arterial EC polarity[53–55]. *Esm1-CreERT2*-mediated loss of *Itgb1* has no impact on embryonic weight, mesenteric artery density, or organization of the intestinal villus vasculature, despite the confirmed loss of integrin β1 protein in GFP+ cells (Fig. 5b, c; Supplementary Fig. 11b–d). However, *Itgb1^iEsm1KO* embryos show a reduction of mesenteric artery diameter relative to control or heterozygous mutant (*Itgb1^iEsm1HET*) littermates (Fig. 5b, c). High-magnification analysis of the large mesenteric arteries reveals no change in the density of EC nuclei but confirms the decrease in vascular diameter (Fig. 5d–f). The inclusion of a *R26-mTmG* reporter allele shows that GFP+ ECs in *Itgb1^iEsm1KO* mesenteric arteries acquire a rounded shape, which is distinct from the strongly elongated appearance of control arterial ECs (Fig. 5d–f). Similar alterations in GFP+ cell morphology are evident in arteries of the intestinal submucosa (Fig. 5g).

We then asked if these observations could be recapitulated in the retinal sprouting angiogenesis model (Supplementary Fig. 12). *Esm1-CreERT2*-mediated loss of *Itgb1* at P6 results in defective extension of tip cells in the *Itgb1^iEsm1KO* mutants compared to *Itgb1^iEsm1HET* control littermates, resulting in blunt areas in the vascular plexus growing front (Supplementary Fig. 12a, b). Loss of *Itgb1* results in reduced Esm1+ cell-derived contribution to the retinal vasculature, affecting overal vascular growth (Supplementary Fig. 12a, b). These results show that despite similar genotypes, the loss of *Itgb1* in Esm1+ cells lead to the different outcomes, due to varying mechanisms and microenvironments, which generates different phenotypic manifestations and defects. Within the retina, Esm1+ cells support sprouting angiogenesis, whereas the arterial progenitor cells in the embryonic gut can enter large mesenteric arteries but require the integrin subunit for their proper integration into the endothelial monolayer (Fig. 5h).

### Artery-derived VEGF-C promotes artery development

In the adult intestine, the VEGFR3 ligand VEGF-C is expressed by vascular and intestinal smooth muscle cells (SMCs), macrophages, and a subset of villus fibroblasts[56–58]. Recent work, supported by scRNA-seq analysis, has identified mesenteric arteries as a major source of VEGF-C with a crucial role in the development of a secondary postnatal lymphatic capillary network[59,60]. In our scRNA-seq data, *Flt4/Vegfr3* is enriched in lymphatic ECs but also in Esm1+ capillary ECs, whereas *Vegfc* is strongly expressed in the arterial endothelium, which is supported by pseudotime analysis (Fig. 6a, b). RNAscope analysis on mesenteric and intestinal cryosections confirms the pattern of *Vegfc* mRNA expression. Despite strong autofluorescence from red blood cells (RBCs), *Vegfc* transcripts are predominantly detected in the mesenteric and submucosal arteries as well as in SOX17+ villus capillaries (Fig. 6c, d). In the intestine, *Vegfc* mRNA is detected throughout the mucosal tissue, suggesting that various embryonic intestinal cell types express this growth factor (Fig. 6d).

To address whether arterial VEGF-C might facilitate the recruitment of Esm1+ cell progeny into the mesenteric vasculature, we introduced the *Bmx-CreERT2* line into a background of mice carrying conditional *Vegfc^lox/lox* alleles[61] (Supplementary Fig. 13a). We performed short-term 4-OHT-induced *Vegfc* inactivation starting from E14.5. After 4 days of *Vegfc* deletion from arteries, the lymphatic vessel area is significantly reduced in *Vegfc^iBmxKO* mutants compared to control littermate embryos (Fig. 6e, h). These results confirm prior studies indicating that arteries are a key source of *Vegfc* for mesenteric lymphangiogenesis[60]. While large mesenteric arteries remain unaffected by the loss of VEGF-C, *Vegfc^iBmxKO* arteries in the intestinal mucosa are significantly thinner (Fig. 6f–h), indicating that arteries closest to the intestine are most affected.

RNAscope analysis confirms the loss of *Vegfc* mRNA expression in large *Vegfc^iBmxKO* mesenteric and submucosal arteries (Supplementary Fig. 13a–c). However, residual *Vegfc* mRNA is still detected in the intestinal mucosa and SOX17+ ECs in *Vegfc^iBmxKO* intestinal villi, which may reflect incomplete *Bmx-CreERT2*-mediated recombination or *Vegfc* expression prior to upregulation of *Bmx*. The latter is consistent with the pseudotime analysis of our scRNA-seq data, showing that *Vegfc* expression is initiated in Esm1+ ECs during early steps of arterial specification and prior to *Bmx* expression (Supplementary Fig. 13d). Nevertheless, analysis of intestines shows that *Vegfc^iBmxKO* villus capillaries show an increase in CD31+ ERG+ ECs area but significantly thinner SOX17+ arterial branches (Fig. 6i, j). These results suggest that arterial *Vegfc* expression promotes the translocation of arterial progenitor cells from intestinal villi into the adjacent submucosal arteries. Accordingly, EC density is increased in *Vegfc^iBmxKO* villi, whereas the diameter of the submucosal arterial network is decreased (Fig. 6g–j).

### VEGFR3 signaling promotes the translocation of Esm1+ cell progeny into arteries

Based on the results above, we hypothesized that signaling by VEGF-C and VEGFR3 enhances the efficient translocation of intestinal arterial progenitor cells into the large mesenteric arteries. To examine the role of the VEGFR3 tyrosine kinase receptor, we inhibited its activity by administering MAZ51 to pregnant females during a short lineage tracing experiment of Esm1+ cell progeny (Fig. 7a). MAZ51 is a potent inhibitor of VEGFR3 kinase activity, which only weakly affects the related receptor tyrosine kinase VEGFR2[62,63]. After MAZ51 treatment, growing LYVE1+ lymphatic capillaries in the intestine are significantly reduced compared to vehicle controls (Fig. 7b–d), indicating efficient inhibition of VEGFR3 activity (Fig. 7d). In contrast, MAZ51 treatment does not affect the villus capillary network, *Esm1-CreERT2*-controlled GFP expression, or SOX17 signal (Fig. 7e, f). Strikingly, the contribution of GFP+ Esm1+ cell progeny to the submucosal and large mesenteric arteries is significantly reduced by MAZ51 relative to vehicle controls (Fig. 7g–i), which is not a result of increased cell apoptosis induced by the MAZ51 treatment (Supplementary Fig. 14a, b). These findings support that inhibiting VEGFR3 activity in arterial progenitor cells impairs their translocation into submucosal and mesenteric arteries.

To test this hypothesis further, we generated mutant embryos carrying loxP-flanked versions of *Flt4* (*Flt4^lox/lox*)[64] together with *Esm1-CreERT2* and *Rosa26-mTmG* alleles. Following tamoxifen administration from E10.5 onward, the appearances of control, *Flt4^iEsm1HET*, and *Flt4^iEsm1KO* embryos are indistinguishable at E18.0 (Supplementary Fig. 15a, data not shown). Immunostaining confirms the expected reduction of VEGFR3 expression in the *Flt4^iEsm1HET* and, more completely, in the *Flt4^iEsm1KO* capillary network at the villus apex compared to control littermates (Supplementary Fig. 15b). Quantification of VEGFR3 shows substantial residual VEGFR3 signal in the *Flt4^iEsm1HET* and *Flt4^iEsm1KO* villus capillary networks, but with significant loss in GFP+ areas (Supplementary Fig. 15b, c). Despite the deletion of VEGFR3 in the GFP+ ECs in *Flt4^iEsm1KO* samples, these results reflect that the villus apex is not entirely composed of *Esm1-CreERT2*-targeted ECs, so that only a subset of ECs in the villus apex loses VEGFR3 (Supplementary Fig. 15b, c).

Unlike MAZ51 treatment, targeted *Flt4* deletion in Esm1-derived cells does not impact lacteal length, suggesting that this approach does not interfere with lymphatic growth (Supplementary Fig. 15b, d). *Flt4^iEsm1KO* mutants exhibit a slight increase in blood vessel width relative to control littermates (Supplementary Fig. 15b, d). Despite the incomplete deletion of VEGFR3 in the villus vasculature, *Flt4^iEsm1KO* mesenteric samples relative to *Flt4^iEsm1HET* littermates show reduced incorporation of GFP+ cells into submucosal arteries and large mesenteric arteries, whereas control (*Flt4^iEsm1WT*) samples show significantly higher incorporation of GFP+ cells (Supplementary Fig. 15e, f).

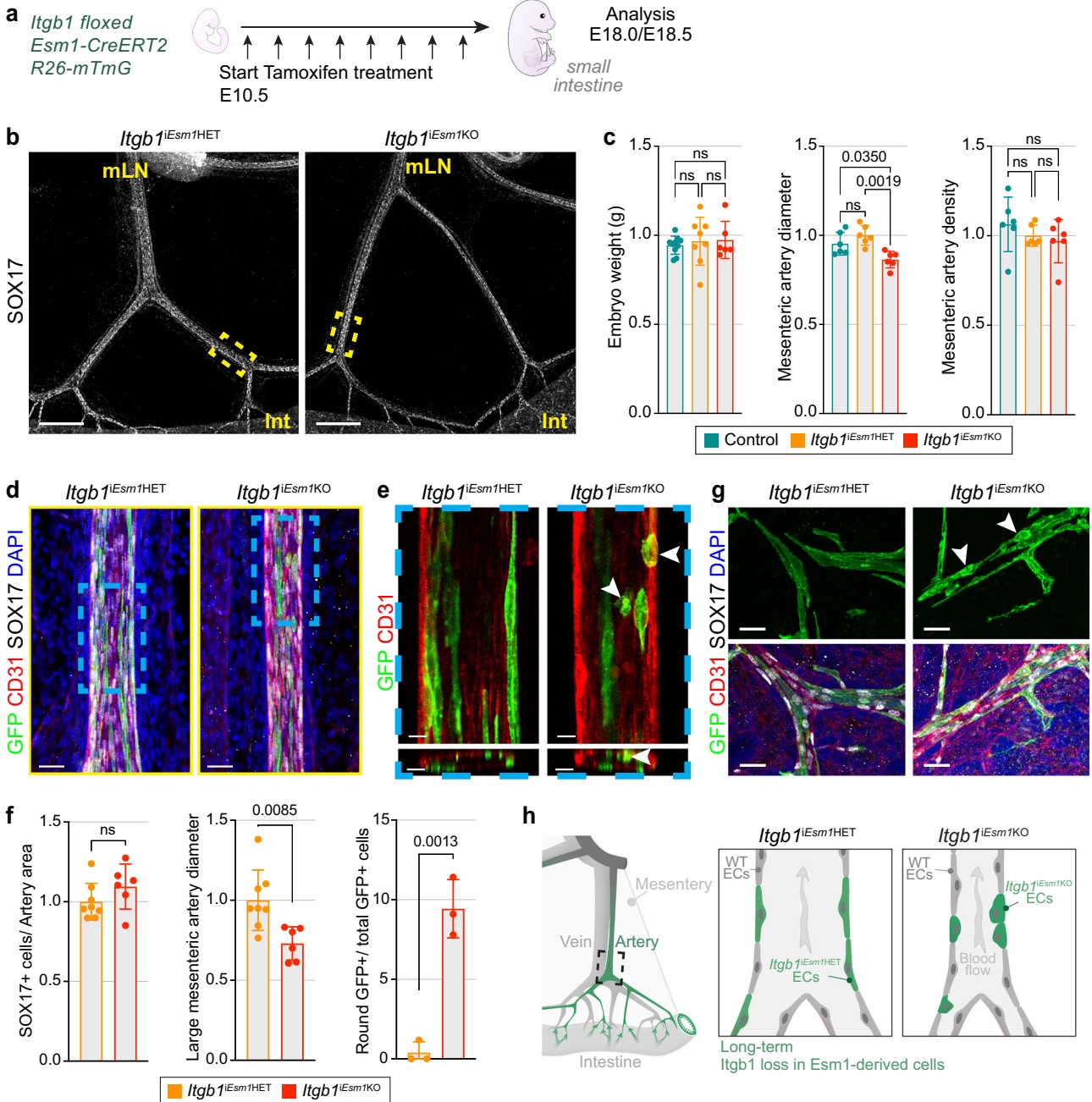

**Fig. 5 | Loss of integrin β1 in Esm1⁺ ECs leads to thinning of mesenteric arteries.** **a** Experimental design of *Esm1-CreERT2*-mediated *Itgb1* inactivation. **b** Whole-mount SOX17 (white) staining of E18.5 mesenteries shows that *Itgb1* deficiency in Esm1⁺ cells (*Itgb1^iEsm1KO^*) reduces average mesenteric artery diameter. Mesenteric lymph node (mLN) and intestine (Int) are indicated. *Itgb1^iEsm1HET^* $n = 3$ and *Itgb1^iEsm1KO^* $n = 3$. Scale bar, 400 μm. **c** Graphs show quantification of embryonic weight, mesenteric artery diameter (in μm, artery area normalized to artery length) and mesenteric artery density (in μm, artery length normalized to tissue area) in control, *Itgb1^iEsm1KO^* mutants and *Itgb1^iEsm1HET^* heterozygous littermates. *Itgb1^iEsm1HET^* group set to 1. Data from 3 experiments were pooled. P values, 1-way ANOVA with Tukey post-hoc test; Error bars, Mean ± SD. Whole-mount staining of mesenteric artery image (**d**) and high magnification longitudinal and transverse single confocal planes (**e**). GFP (green), CD31 (red), SOX17 (white) and DAPI (blue). White arrowheads indicate rounded GFP⁺ cells in *Itgb1^iEsm1KO^* mutant. *Itgb1^iEsm1HET^* $n = 3$; *Itgb1^iEsm1KO^* $n = 3$. Scale bars, 30 and 20 μm. **f** Graphs show quantification of SOX17⁺ cells in large mesenteric arteries (normalized to artery area), average diameter of large mesenteric arteries (in μm, artery area normalized to artery length) and number of round GFP⁺ cells in large mesenteric arteries (normalized to total number of GFP⁺ cells). P values, 2-tailed unpaired Student's *t* test; Error bars, Mean ± SD. **g** Whole-mount staining of E18.5 intestine shows that *Itgb1* inactivation in Esm1⁺ ECs alters the shape of GFP⁺ cells (green) in intestinal submucosal arteries. CD31 (red), SOX17 (white) and DAPI (blue). White arrowheads indicate rounded GFP⁺ cells. *Itgb1^iEsm1HET^* $n = 3$; *Itgb1^iEsm1KO^* $n = 3$. Scale bar, 30 μm. **h** Long-term tamoxifen treatment using *Esm1-CreERT2* mouse line generates mosaic mesenteric arteries composed of intestinal Esm1-derived ECs (Green cells). Long-term *Itgb1* deletion in Esm1-derived cells (*Itgb1^iEsm1KO^*) leads to cell rounding and reduced mesenteric artery diameter.

To inactivate *Flt4* more efficiently in villus ECs, we made use of the finding that *Aplnr-CreERT2*-labeled cells give rise to Esm1⁺ ECs and, subsequently, arterial endothelium. We therefore introduced the *Aplnr-CreERT2* line into the *Flt4^lox/lox^* background. To avoid lymphatic involvement (Supplementary Fig. 4h–j), treatment was initiated at E13.5 (Fig. 8a). By E18.0, we observed significant loss of VEGFR3 specifically in *Flt4^iAplnrKO^* blood vessels without affecting the intestinal (lymphatic) lacteals (Fig. 8b, c). Interestingly, the *Flt4^iAplnrKO^* villus blood

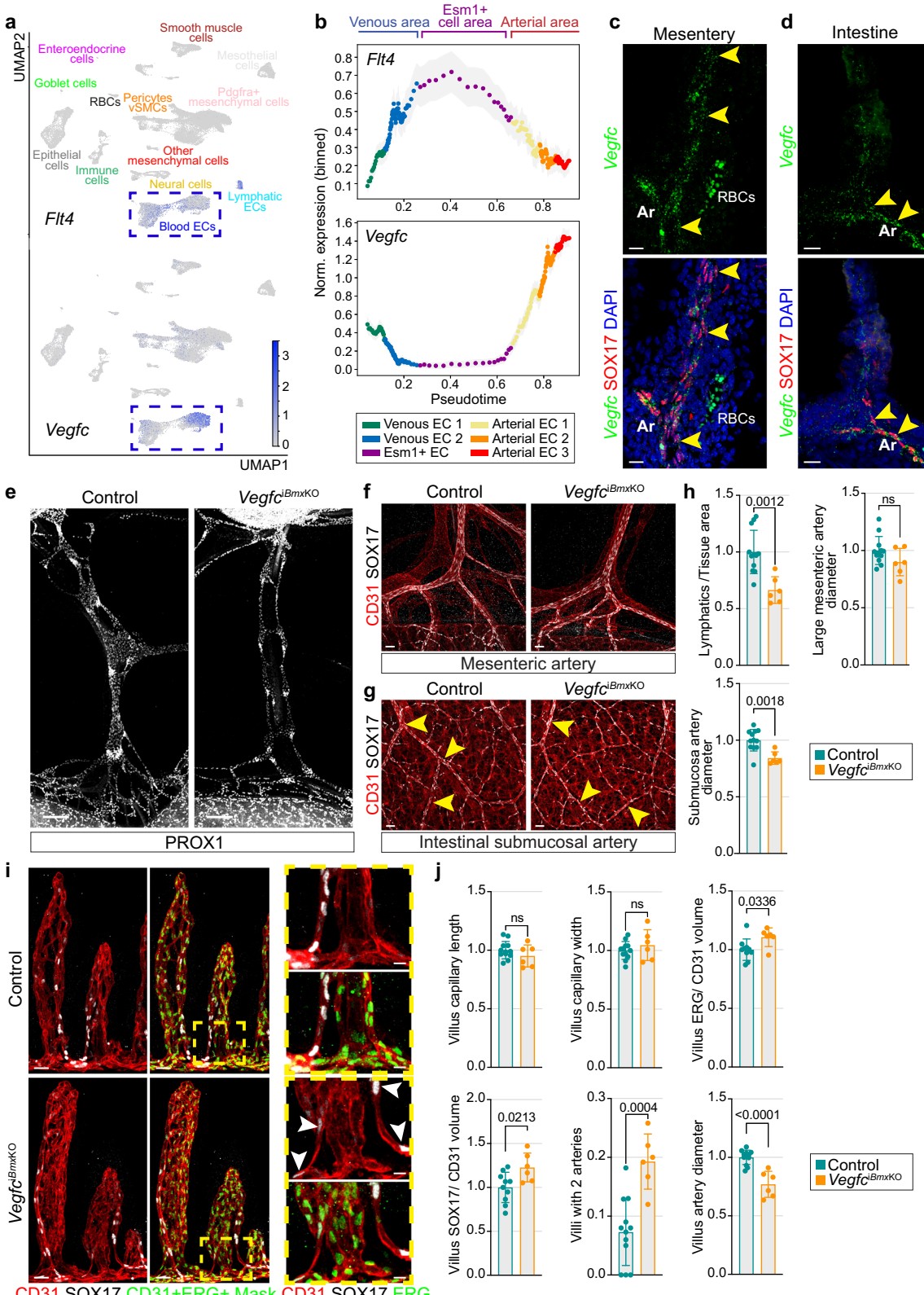

vessel network is reduced, whereas VEGFR3[high] lacteals grow longer, which may reflect higher availability of VEGF-C due to the lack of VEGFR3 in blood vessel ECs. Upstream of the villus, submucosal arteries are notably thinner than in control littermates (Fig. 8d, e), mirroring the phenotype observed in *Vegfc*[iBmxKO] mutants (Fig. 6g, h). Together with the effects seen following MAZ51 treatment and *Vegfc* and *Flt4* inactivation, these results indicate that VEGFR3 promotes

the recruitment of Esm1+ ECs from villus capillaries into upstream arteries.

## Discussion

Arteries are an essential part of the vascular tree and arterial malfunction is causally linked to detrimental, sometimes life-threatening human diseases. Mesenteric arteries carry blood from the abdominal

**Fig. 6 | Loss of arterial *Vegfc* impairs arterial and lymphatic vessel development. a** UMAP plots showing the expression of *Flt4* and *Vegfc* in all cells isolated from intestine and mesentery. (**b**) Pseudotime plots showing increased *Flt4* expression level in the Esm1[+] EC area, whereas *Vegfc* decreases relative to the artery area. RNAscope detects *Vegfc* mRNAs (green, arrowheads) in SOX17[+] (red) mesenteric arteries (**c**) and in SOX17[+] intestinal arteries (**d**) at E19.0. Nuclei, DAPI (blue). Arteries (Ar) and red blood cells (RBCs), a source of background signal, are indicated. *n* = 5 each. Scale bar, 15 μm. **e** Loss of *Vegfc* from E14.5 in Bmx[+] arteries impairs mesenteric lymphatic vessel (PROX1, white) development at E19.0. Control *n* = 12; *Vegfc*[iBmxKO] *n* = 6. Scale bar, 250 μm. Loss of *Vegfc* from E14.5 in BMX[+] arteries does not affect large SOX17[+] (white) mesenteric arteries (**f**) but leads to thinning of arteries (arrowheads) in the intestinal mucosa (**g**) at E19.0. CD31 (red). Control *n* = 12; *Vegfc*[iBmxKO] *n* = 6. Scale bars, 50 μm. **h** Quantification of mesenteric lymphatic vessel area (μm², normalized to tissue area), large mesenteric artery (close to the

intestine) and intestinal submucosal artery diameter (in μm, artery area normalized to artery length) in *Vegfc*[iBmxKO] E19.0 embryos and control littermates. P values, 2-tailed unpaired Student's *t* test; Error bars, Mean ± SD. **i** Whole-mount staining showing ERG (green), CD31 (red) and SOX17 (white) staining of control and *Vegfc*[iBmxKO] E19.0 intestinal villi. Panels on right show higher magnifcations of boxed areas. A mask was applied to identify CD31[+] ERG[+] cells (green, left panels). Arrowheads indicate ERG[+] SOX17[+] villus arteries. Control *n* = 12; *Vegfc*[iBmxKO] *n* = 6. Scale bars, 30 and 10 μm. **j** Quantification of villus blood capillary network length and width (μm), total villus vessel nuclear density (μm³, total ERG[+] volume normalized to CD31[+] volume), SOX17[+] nuclear density (μm³, total SOX17[+] volume normalized to CD31[+] volume), number of villi with more than one main CD31[+] SOX17[+] vessel branch, and average villus SOX17[+] vessel diameter (μm). For all analysis, P values were determined by 2-tailed unpaired Student's *t* test; Error bars, Mean ± SD.

aorta to the gastrointestinal tract and are thereby indispensable for the function of this organ. Accordingly, the blockade or narrowing of mesenteric arteries, which occurs in human patients due to the build-up of atherosclerotic plaques or other reasons, can lead to severe abdominal pain, intestinal damage and the need for surgical intervention[65]. Our study provides fundamental insights into the development of arteries of the mesentery and intestinal wall. In particular, we show that a small population of Esm1[+] cells, located inside intestinal villi, gives rise to arterial ECs both in the intestine and mesentery. Accordingly, the formation of these arteries requires EC migration over substantial distances in addition to arterial specification. Previous work has established that Esm1[+] tip cells in the postnatal retina, which are located at the distal end of endothelial sprouts and thereby at the border to avascular tissue, generate arterial but also capillary ECs[12,13]. Remarkably, Esm1[+] cells in the embryonic intestine express typical tip cell markers even though they are part of patent capillary tubes and do not show sprouting behavior. Thus, Esm1/ endocan expression in this setting is likely to reflect elevated levels of growth factor signaling, especially of VEGF-A and VEGF-C, as has been shown previously[66,67]. Mechanistically, Esm1 has been shown to bind to fibronectin and displace fibronectin-bound VEGF-A, thereby increasing the bioavailability and signaling capacity of the growth factor[31]. As a result, Esm1 enhances endothelial sprouting activity as well as vascular permeability[31,68]. This role of Esm1 might be also relevant in the apex of the villus vasculature where VEGF-A modulates EC junctions to enhance nutrient uptake in the adult murine intestine[24]. In the context of arterial development, Esm1 might facilitate the migration of pre-arterial ECs into the arterial walls and further into mesenteric arteries. Apart from VEGF-A, which is highly expressed in the distal villus, the related ligand VEGF-C is also relevant. As previously described for the murine postnatal and adult intestine[24,59,60], VEGFR3 expression marks Esm1[+] cells in the embryonic intestine, whereas its ligand, VEGF-C, is provided by submucosal and mesenteric arteries. Functionally, the migration and arterial incorporation of Esm1[+] (and VEGFR3[+]) ECs is compromised after genetic or pharmacological disruption of signaling by VEGF-C and its receptor VEGFR3. The function of VEGFR3 and VEGF-C is known to be crucial for the lymphatic vessel development and maintenance, but also modulates sprouting angiogenesis and blood vessel growth in mouse and zebrafish models[69–74]. Our new findings reveal an additional and unexpected function of this ligand-receptor pair, namely the guidance of Esm1[+] ECs into the growing arterial vasculature.

Our genetic fate tracking experiments show that Esm1[+] cell progeny inside arteries acquires the typical elongated shape characteristic of arterial ECs[75]. *Esm1-CreERT2*-mediated inactivation of integrin β1, an important subunit of many integrin receptor heterodimers, leads to the rounding of Esm1[+] cell progeny and reduces the diameter of embryonic mesenteric arteries. This rounding phenotype is reminiscent of previous findings showing that pan-endothelial (but incomplete and therefore mosaic) inactivation of the *Itgb1* gene impairs

arteriolar lumen formation due to defective EC polarization[55]. Other studies showed that integrin β1 is also required for endothelial sprouting in the postnatal retina, arteriole formation in the ischemic heart but also for the barrier function of the endothelium and the prevention of vascular leakage[54,76,77]. All these aspects might be of relevance for the formation and function of the arterial network in the intestine and mesentery.

Our findings also raise the question whether the recruitment of Esm1[+] cell progeny is the predominant or even sole process responsible for mesenteric artery development in the embryo. The existence of alternative, redundantly acting mechanisms might contribute to biological robustness and resilience. In the developing heart, ECs derived both from the sinus venosus and the endocardium contribute to coronary blood vessel formation, and both pools of arterial progenitors can compensate for each other[6,78–80]. Similarly, multiple non-venous sources of lymphatic ECs have been identified in various organs, including the mesentery[36,81]. Our own findings show that fully differentiated mesenteric arterial ECs overcome the cell cycle arrest that is characteristic for ECs undergoing arterial differentiation[13–15] and exhibit some level of proliferation. Similarly, EC proliferation in situ contributes to the regeneration of damaged aortic endothelium in the adult mouse[82]. Thus, it is feasible that the expansion of mature arterial ECs in the mesentery and intestinal wall might be able to compensate for insufficient specification of prearterial cells. Alternatively, processes such as vessel remodeling and pruning[18,83], which play important roles during the angiogenic expansion of the vasculature, might help to ensure or restore sufficient arterial blood flow. Taken together, our findings establish a fundamental framework for artery development in the intestine and mesentery. We propose that intestinal villi represent a niche microenvironment for the induction of Esm1[+] arterial progenitors, which will integrate into growing arteries, but not into veins or lymphatic vessels, and thereby contribute to the expansion of the arterial network.

## Methods
### Mouse models
All animal experiments were performed according to the institutional guidelines and laws, approved by local animal ethical committee and were conducted at the Max Planck Institute for Molecular Biomedicine with necessary permissions (Az 81-02.04.2019.A114) granted by the Landesamt für Natur, Umwelt und Verbraucherschutz (LANUV) of North Rhine-Westphalia, Germany.

*Esm1-CreERT2*[31], *Bmx-CreERT2*[33], *Aplnr-CreERT2*[32], *Itgb1*[lox/lox84], *Flt4*[lox/lox64], *Vegfc*[lox/lox61], *R26-mTmG*[30], *Ai14*[85], *Cdh5-mTnG*[86] and *Hey1-eGFP (Tg(Hey1-EGFP)ID40Gsat;* http://www.gensat.org) were previously described.

### Embryonic, postnatal and adult mouse treatments
For embryo experiments, vaginal plugs were verified in the morning. Embryonic age (E) was determined according to the day of the

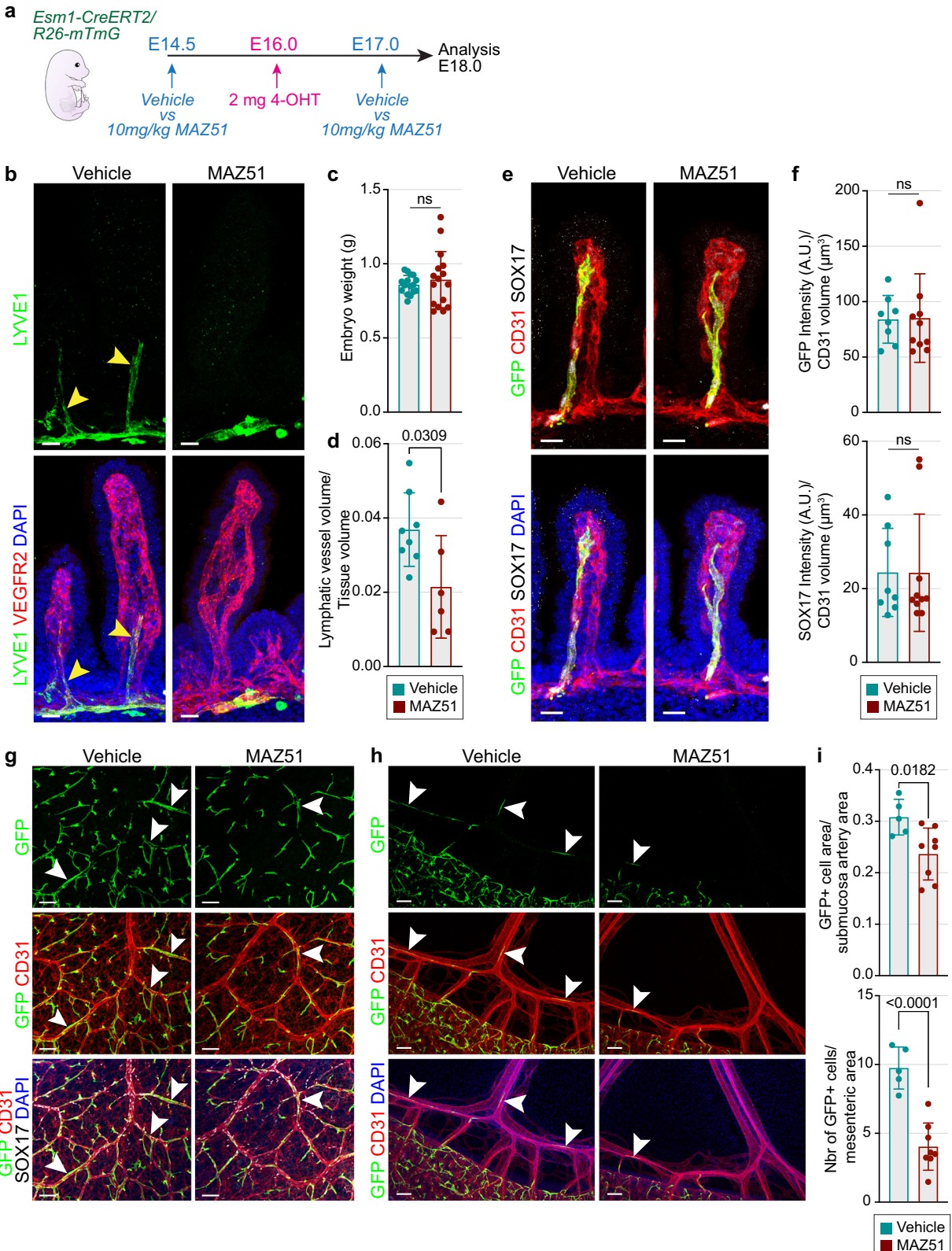

vaginal plug (E0.5). For lineage tracing embryo experiments, one injection of 2 mg total 4-hydroxytamoxifen (4-OHT; Sigma, #H7904) in oil was injected intraperitoneally (i.p.) in pregnant mice. For lineage tracing experiments with VEGFR3 tyrosine kinase inhibition, MAZ51 (2% DMSO in PBS, #HY-116624) was injected subcutaneously (s.c.) at E14.5 and E17.0. One dose of 2 mg of 4-OHT in oil was injected i.p. in pregnant mice at E16.0. Embryos were collected 48 h later at E18.0. Click-it EdU cell proliferation

Alexa Fluor 647 kit (Life Technologies, #C10340) was used for EdU detection. When indicated, 5 µg/g EdU was injected i.p. 1 h before sacrifice.

For continuous Esm1+ cell tracking, pregnant females were injected with 2.5 mg tamoxifen (Sigma, #T5648) and progesterone (Sigma, #P3972) mixed in oil s.c. starting from E10.5 every day until the day before embryo collection. *Esm1-CreERT2*tg/+ *R26-mTmG*tg/+ embryos were analyzed at E13.5 and E16.5.

**Fig. 7 | VEGFR3 inhibition reduces Esm1⁺ cell contribution to mesenteric arteries. a** Experimental design of short-term 48 h lineage tracing experiment with MAZ51 or vehicle treatment. **b** Whole-mount images of E18.0 intestine shows regression of LYVE1⁺ (green) lymphatic vessels after MAZ51 treatment. VEGFR2 (red) and DAPI (blue). Arrowheads mark LYVE1⁺ lacteals. Vehicle *n* = 8; MAZ51 *n* = 6. Scale bar, 20 μm. **c** Embryonic weight is comparable between vehicle and MAZ51 groups. P value, Welch's *t* test; Error bars, Mean ± SD. **d** Quantification of LYVE1⁺ lymphatic capillaries in E18.0 intestine (μm³, normalized to tissue volume). P value, 2-tailed unpaired Student's *t* test; Error bars, Mean ± SD. **e** Whole-mount of E18.0 intestine at 48 h after *Esm1-CreERT2*-mediated reporter activation. GFP (green),

CD31 (red), SOX17 (white) and DAPI (blue). Vehicle *n* = 8; MAZ51 *n* = 10. Scale bar, 20 μm. **f** Quantification of average GFP and SOX17 intensity in villus capillaries (per μm³). P values, 2-tailed unpaired Student's *t* test; Error bars, Mean ± SD. Whole-mount of E18.0 intestines (**g**) and mesenteries (**h**) showing decreased GFP⁺ cell contribution (arrowheads) to arteries after MAZ51 treatment. GFP (green), CD31 (red), SOX17 (white) and DAPI (blue). Vehicle *n* = 5; MAZ51 *n* = 8. Scale bar, 80 μm and 100 μm. **i** Quantification of GFP⁺ area in submucosal arteries (μm², normalized to total arterial area) and number of GFP⁺ cells found in mesenteric tissue (normalized to total tissue area). P values, 2-tailed unpaired Student's *t* test; Error bars, Mean ± SD.

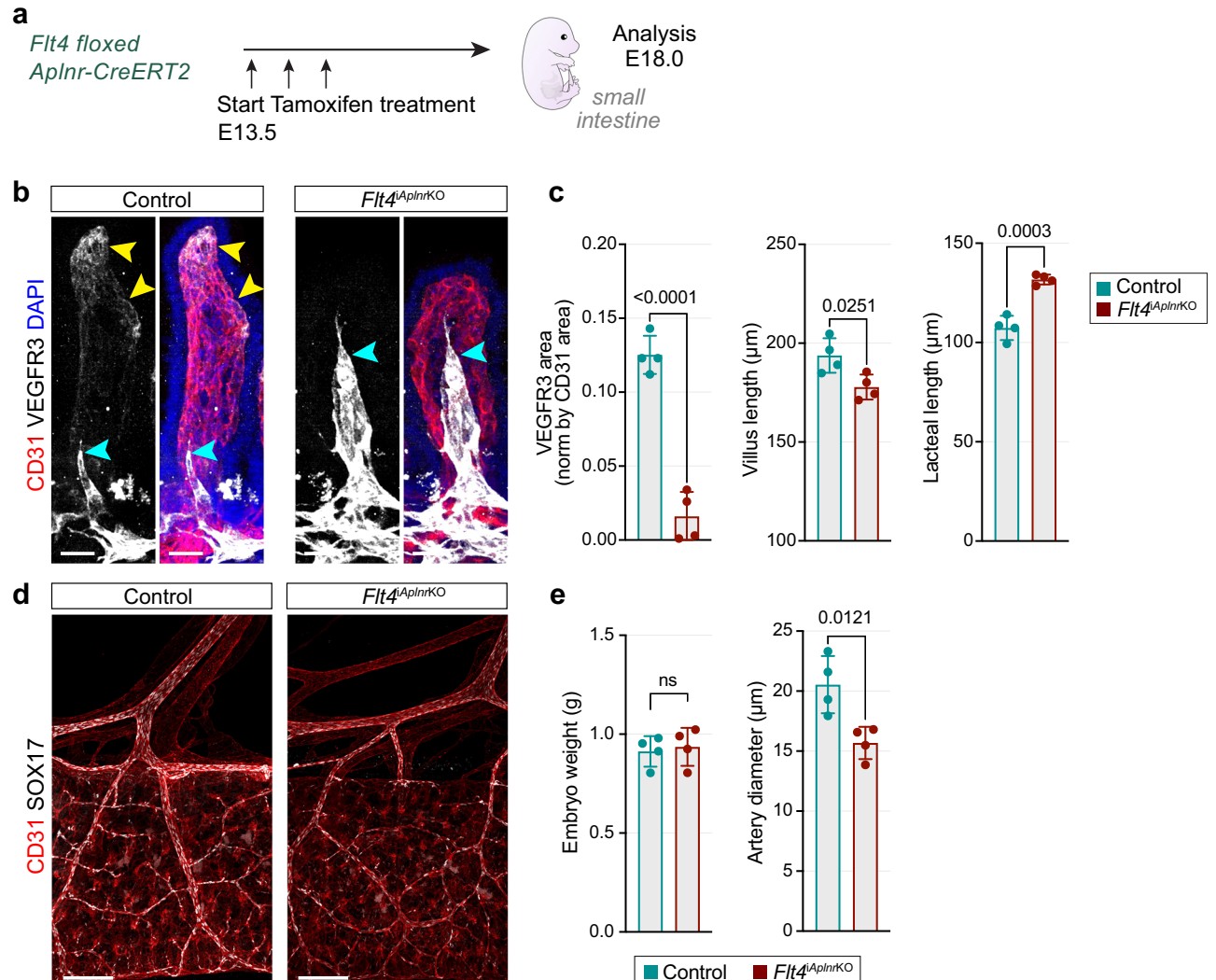

**Fig. 8 | Loss of *Flt4* in Aplnr⁺ ECs impairs artery development. a** Experimental design for Aplnr⁺ cell-specific *Flt4* inactivation from E13.5. **b** Loss of VEGFR3 (white) expression, reduced villus blood capillary length and increase lacteal length in *Flt4^iAplnrKO* mutant villi compared to control littermates at E18.0. VEGFR3 (white), CD31 (red) and DAPI (blue). Yellow arrowheads mark VEGFR3⁺ capillaries in villus apex. Blue arrowheads mark VEGFR3⁺ lacteals. Control *n* = 4; *Flt4^iAplnrKO* *n* = 4. Scale bar, 30 μm. **c** Quantification of VEGFR3⁺ area in villus blood vessels (μm², normalized to vessel area), villus capillary network length (μm) and villus lacteal length

(μm). P values, 2-tailed unpaired Student's *t* test; Error bars, Mean ± SD. **d** Whole-mount of E18.0 intestine and mesentery showing thinner SOX17⁺ arteries in *Flt4^iAplnrKO* mutants compared to control littermates. CD31 (red) and SOX17 (white). Control *n* = 4; *Flt4^iAplnrKO* *n* = 4. Scale bar, 200 μm. **e** Embryonic weight is comparable between control and *Flt4^iAplnrKO* littermates. Average artery diameter leaving the intestine in *Flt4^iAplnrKO* littermates is reduced compared to controls. P value, 2-tailed unpaired Student's *t* test; Error bars, Mean ± SD.

In loss-of-function experiments, *Itgb1^iEsm1KO* embryos (*Itgb1^lox/lox Esm1-CreERT2^tg/+*) were compared to *Itgb1^iEsm1HET* (*Itgb1^lox/+ Esm1-CreERT2^tg/+*) or control littermates (*Itgb1^+/+; Esm1-CreERT2^tg/+* or mice without the *Esm1-CreERT2* transgene). All embryos carried the reporter transgene *R26-mTmG* (*R26-mTmG^tg/+*). In *Esm1-CreERTR2*-controlled *Flt4* loss-of-function experiments, *Flt4^iEsm1KO* embryos (*Flt4^lox/lox Esm1-*

*CreERT2^tg/+*) were compared to *Flt4^iEsm1HET* (*Flt4^lox/+ Esm1-CreERT2^tg/+*), *Flt4^iEsm1WT* (*Flt4^+/+ Esm1-CreERT2^tg/+*), or control littermates without the *Esm1-CreERT2* transgene. All embryos carried the reporter transgene *R26-mTmG* (*R26-mTmG^tg/+*) and some carried the *Ai14* transgene. *Itgb1* and *Flt4* long-term deletion experiments were induced with 2.5 mg tamoxifen and progesterone mixed in oil s.c. starting from E10.5 every

day until the day before embryo collection. For *Aplnr-CreERT2*-controlled *Flt4* loss-of-function experiments, *Flt4*[iAplnrKO] (*Flt4*[lox/lox] *Aplnr-CreERT2*[tg/+]) embryos were compared with control littermates (*Flt4*[lox/lox] *Aplnr-CreERT2*[+/+]). Long-term deletion experiments were induced with 2.5 mg tamoxifen and progesterone mixed in oil s.c. at E13.5, E14.5 and E15.5. For *Bmx-CreERT2*-controlled *Vegfc* loss-of-function experiments, *Vegfc*[iBmxKO] (*Vegfc*[lox/lox] *Bmx-CreERT2*[tg/+]) embryos were compared with control littermates (*Vegfc*[lox/lox] *Bmx-CreERT2*[+/+]). Short-term deletion experiments were induced with 2 mg of 4-OHT in oil injected s.c. starting from E14.5 every day until the day before embryo collection.

For postnatal lineage tracing experiments, *Esm1-CreERT2*[tg/+] *R26-mTmG*[tg/+] pups were injected once at P1 with 50 μg 4-OHT in oil s.c. and analyzed at P8 or P21. *Itgb1* deletion experiments were induced with 50 μg tamoxifen mixed in oil at P1, P2 and P3.

For lineage tracing in adult mice, 20-24 week-old *Esm1-CreERT2*[tg/+] *R26-mTmG*[tg/+] males were injected once with 2 mg 4-OHT in oil i.p. and analyzed 24 h or 2 weeks later.

## Mouse embryonic mesenteric and intestinal tissue collection and staining procedures

Embryonic gastro-intestinal tracts were collected in ice-cold PBS. Duodenum (around 1 cm at E18.0) and first part of jejunum (next 1 cm) were collected, cut open and pinned in a 12-well elastomer-coated dish and thoroughly washed with ice-cold PBS. From the remaining tissue, intestine was pinned in a circle to visualize the mesentery. Clean duodenum, jejunum and mesentery were fixed with ice-cold 4% paraformaldehyde (PFA) in PBS (Sigma, #P6148) overnight on a rotating platform at 4 °C. Samples were then washed with ice-cold PBS and incubated overnight with 10% sucrose in PBS followed by 20% sucrose and 10% glycerol in PBS. For staining, 0.5 cm-long intestine pieces or mesenteric tissues were washed with PBS, permeabilized with 0.5% Triton X-100 and blocked with 5% donkey serum overnight. Tissues were incubated with primary, followed by secondary antibodies overnight on a rotating platform at 4 °C. Intestine samples were incubated overnight with Histodenz (Sigma, #D2158) at room temperature (RT). All samples were mounted in Histodenz with DAPI (Sigma, #D9542). A list of antibodies used in this study is provided in Supplementary Table 1.

## Mouse postnatal mesentery, postnatal and adult intestine collection and staining

The protocol was adapted from previous publications[87,88]. Pups and adult mice were perfused with PBS then 4% PFA in PBS after anesthesia. Isolated tissues were washed in ice-cold PBS and duodenum and first jejunum parts were collected, cut open and pinned in a 6-well or 10 cm elastomer-coated dish and thoroughly washed with ice-cold PBS. The remaining postnatal jejunum and ileum were pinned down in a circle to visualize the mesentery. Clean duodenum and jejunum samples and mesenteries were fixed with ice-cold 4% PFA in PBS overnight on a rotating platform at 4 °C. Samples were then washed with ice-cold PBS and incubated overnight with 10% sucrose in PBS, followed by 20% sucrose and 10% glycerol in PBS. For staining, 1 cm-long intestinal pieces and mesenteries were washed with PBS, permeabilized with 0.5% Triton X-100 and blocked with 5% donkey serum overnight. Tissues were incubated with primary, followed by secondary antibodies overnight on a rotating platform at 4 °C. Intestine samples were incubated overnight with Histodenz at RT. All samples were mounted in Histodenz with DAPI. A list of antibodies used in this study is provided in Supplementary Table 1.

## Adult mouse retina collection and staining

Tissues were fixed with 4% PFA for 1 h at RT, washed with ice-cold PBS and permeabilized with 0.5% Triton X-100 and blocked with 5% donkey serum overnight on a rotating platform at 4 °C. Tissues were incubated with primary antibodies followed by secondary antibodies overnight on a rotating platform at 4 °C. Retinas were mounted in Fluoromount-G (Southern Biotech, #0100-01) with DAPI.

## Postnatal P5 mouse retina collection and staining

Retina immunostaining was performed as previously described with minor modifications[89]. Tissues were collected and fixed in 4% PFA in PBS. Dissected retinas were incubated in blocking buffer (1% BSA, 0.3% Triton X-100 in PBS) for 2 h at 4 °C, rinsed with modified Pblec buffer (1 mM CaCl2, 1 mM MgCl2, 0.1 mM MnCl2, 0.1% Triton X-100 in PBS), and incubated with primary antibodies diluted in modified Pblec buffer overnight at 4 °C. After washing in blocking buffer diluted 1:1 with PBS and 3 washes in PBS, retinas were incubated for 1 h at RT with secondary antibodies diluted in blocking buffer. Retinas were then washed, refixed with 4% PFA for 20 min at RT and washed twice with PBS prior to mounting with Fluoromount-G (Southern Biotech, #0100-01).

## Staining of intestine cryosections

Tissues were fixed with 4% PFA in PBS overnight, washed with ice-cold PBS and incubated overnight with 30% sucrose in PBS. Intestines were embedded in OCT (Leica Biosystems, #14020108926) and kept at −80 °C until sectioning. 60 μm cryosections were thawed, fixed with 4% PFA in PBS for 5 min, washed in PBS, permeabilized with 0.5% Triton X-100 in PBS and blocked with 5% donkey serum for 30 min. Samples were incubated with primary antibodies in blocking buffer overnight at 4 °C. Slides were then washed and incubated with secondary antibodies for 1 h at RT. Slides were washed and mounted in Fluoromount-G (Southern Biotech, #0100-01) with DAPI.

## RNAscope analysis of tissue cryosections

Tissues were fixed with 10% Neutral-Buffered Formalin (NBF) for 24 h at RT, washed with ice-cold PBS and incubated overnight with 30% sucrose in PBS. Mesenteries and intestines were embedded in OCT (Leica Biosystems, #14020108926) and kept at −80 °C until sectioning. For *Vegfc* mRNA detection, samples were processed according to the protocol provided by Advanced Cell Diagnostics (ACD, RNAscope Multiplex Fluorescent v2 Assay combined with Immunofluorescence). Briefly, 22 μm cryosections were thawed, fixed with 10% NBF and dehydrated with 50%, 70% and 100% EtOH in distilled water. Samples were then treated with RNAscope hydrogen peroxide and transferred in antigen retrieval buffer before the incubation with primary antibodies overnight at 4 °C. The next day, RNAscope Multiplex fluorescent v2 assay was performed using *Vegfc* and control probes (see Supplementary Table 2). To detect primary antibodies, Alexa-fluor secondary antibodies diluted in co-detection antibody diluent were added for 40 min at RT. Slides were incubated with DAPI and mounted in Fluoromount-G.

## Image acquisition and analysis

All images were captured using Leica SP8 or Zeiss LSM 980 confocal microscopes, and analyzed using Imaris, ImageJ and Photoshop softwares. All images of whole-mount mesenteric staining are shown in the same orientation, i.e. intestine at the bottom of the image.

## Cell isolation and scRNA-seq experiments

For all scRNA-seq experiments, *Esm1-CreERT2*[tg/+] and *R26-mTmG*[tg/+] E18.0 embryos (injected with 2 mg 4-OHT at E17.0) were dissected in ice-cold PBS. The small intestines and mesenteries were separated, and the pancreas and mesenteric lymph node were removed. The small intestines were cut open, and both tissues were thoroughly washed in ice-cold PBS. Mesenteries and intestines were then transferred in FACS complete medium (Phenol red-free DMEM with 5% FBS). Digestion was performed in PBS 0.1% BSA comprising 0.25 mg/ml Liberase DH (Roche, #05401054001) and 0.08 mg/ml DNaseI (Sigma, #DN25) at 37 °C for 30 min for mesenteries and 1 h for intestines. Digestion was

completely stopped by adding FACS complete medium to the cell suspensions. Cells were subsequently filtered through a 40 μm cell strainer (Falcon, #352340) in FACS complete medium and centrifuged for 3 min at 4 °C.

For total mesenteric and small intestine cells, pellets were resuspended in red blood cell (RBC) lysis buffer (Sigma, #R7757) for 3 min at RT, and washed with FACS complete medium. Each cell suspension was directly loaded onto a microwell cartridge of the BD Rhapsody Express system (BD Biosciences, #400000847) and libraries were prepared using the BD Rhapsody WTA Reagent kit (BD Biosciences, #633802) following the manufacturer's instructions. scRNA-seq libraries were evaluated and quantified by Agilent Bioanalyzer using High sensitivity DNA kit (#5067-4626) and Qubit (ThermoFisher Scientific, #Q32851). Individual libraries were pooled, diluted to 4 nM and sequenced by using NextSeq 500/550 High Output kit (150 cycle, Illumina) with a NextSeq500 sequencer (Illumina).

For fluorescence-activated cell sorting (FACS) of intestinal ECs, cell pellets were incubated with Fc block (antibody to CD16/32; BD Biosciences, #553142) for 10 min on ice and stained with conjugated antibodies in FACS buffer for 30 min (see Supplementary Table 3). All GFP+ and GFP− ECs were individually sorted using a FACSAria Fusion (BD Biosciences). GFP+ ECs (around 9000 cells) were complemented with GFP− ECs to reach a total of 40,000 cells and further processed for scRNA-seq library preparation and sequencing as described for total intestinal and mesenteric cells.

### scRNA-seq preprocessing
Raw FASTQ reads were quality and adapter trimmed using TrimGalore! (version 0.6.4 length cutoff 66, quality cutoff 20). The UMI, complex barcode, and sample tags were extracted and demultiplexed using custom scripts (rhapsody-extract-barcode and rhapsody-demultiplex).

Reads were mapped to the GRCm38 reference genome, *Esm1-CreERT2*[g/+], and *R26-mTmG*[g/+] construct sequences with Gencode annotations vM22, using STAR version 2.7.3a[90] (−soloType CB_UMI_Simple −soloCellFilter None −soloFeatures Gene −soloCBstart 1 −soloCBlen 27 −soloUMIstart 28 −soloUMIlen 8 −outFilterMultimapNmax 1 −soloCBwhitelist rhapsody_whitelist.txt).

Raw counts were imported as AnnData[91] objects. We removed low complexity barcodes with the knee plot method, and further filtered out cells with a total contribution above 20% of reads belonging to mitochondrial mRNA. Doublets were predicted with scrublet[92] and cells with a doublet score above 0.1 have been removed. Finally, each sample's gene expression matrix was normalized using scran (1.22.1)[93] with Leiden clustering[94] input at resolution 0.5.

G2/M and S phase scores were assigned to each cell using gene lists from ref.[95] and the scanpy (1.8.2)[96] sc.tl.score_genes_cell_cycle function. In some analyses, the Scanpy sc.pp.regress_out was used to regress out these scores in UMAP and clustering.

Additionally, we used Tricycle (version 1.16.0)[43] to predict cell-cycle phase using "project_cycle_space" with "species = mouse", followed by "estimate_cycle_position". Cells were assigned a cell-cycle phase by Tricycle score: $0.5\pi - \pi = $ S-phase; $\pi - 1.75\pi = $ G2M-phase; other: G1-phase.

### scRNA-seq embedding, clustering and annotation
At this stage, samples were merged. For 2D embedding, the expression matrix was subset to the 2000 most highly variable genes (sc.pp.highly_variable_genes, flavor "seurat"). The top 50 principal components (PCs) were calculated and batch-corrected using Harmony (0.0.5)[97]. The PCs served as basis for k-nearest neighbor calculation (sc.pp.neighbors, n_neighbors=30), which were used as input for UMAP[98] layout (sc.tl.umap, min_dist=0.3).

Known marker genes were plotted using Scanpy (1.7.1) scanpy.pl.dotplot, cell populations were clustered using scanpy.tl.leiden at resolution 0.1 for annotation. The endothelial cell population was subclustered separately at Leiden resolution 0.9, and annotated using known marker genes.

### scRNA-seq trajectory and pseudotime analyses
PAGA[44] from the Scanpy package was used to calculate non-mitotic EC population connectivities and determine trajectories. Based on PAGA and Esm1 lineage tracing information, we calculated diffusion pseudotime[44,99] using scanpy.tl.dpt, choosing the Venous EC1 population as starting cluster.

Expression values for all genes with a minimum average normalised expression larger than 0.3 to reduce noise were binned into 200 bins according to pseudotime. The resulting z-transformed matrix was clustered hierarchically (scipy 1.10.0 scipy.cluster.hierarchical, Ward linkage, Euclidian distance metric) to identify gene expression patterns. Clusters were obtained with a tree distance cutoff of 60 in the dendrogram. Profiles for select clusters were plotted along binned pseudotime, celltype annotations for each bin were determined by majority vote. In profile plots for individual genes, the confidence interval of expression values is plotted as grey background (1.96*SD(expr)/sqrt(n)).

Enrichr[100] was used to calculate Gene Ontoloy (BP, CC) enrichment on the expression profile cluster containing Esm1.

### Integration with P6 retinal endothelial tip cells
For the comparison of Esm1+ cells from the intestinal villus with sprouting retinal tip cells, we reprocessed the publicly available single-cell data from Zarkada et al. 2021[46]. Raw FASTQ files from P6 WT were processed analogously to the intestine and mesentery samples above. Divergent STARsolo options were "−soloType Droplet −soloCBwhitelist 10xv3_whitelist.txt −soloCBlen 16 −soloUMIstart 17 −soloUMIlen 10". Mitochondrial mRNA content cutoff was set to 10%. AnnData objects were merged (outer join) with intestine and mesentery data using Scanpy concatenate. The merged data was reclustered at Leiden resolution 0.05 to identify ECs; the EC population was then further subclustered at resolution 1.0 to identify and remove contaminants and doublets. ECs from the Zarkada et al. data were subclustered at resolution 0.7, and the Esm1+ cluster was identified and annotated accordingly. Differential expression was performed on Esm1+ clusters from the original and Zarkada et al. data using a pseudobulk approach based on pyDESeq2[101].

### Quantifications and statistics
For whole-mount mesenteric artery and lymphatic vessel analysis, mesenteric tissue (DAPI), artery (CD31+ SOX17+) and lymphatic vessel (PROX1+) areas and artery length were quantified using Photoshop and ImageJ softwares. In E18.5 Esm1 lineage tracing experiments, GFP+ cells in the mesenteric arteries were manually counted. In other embryonic mesenteric tissue lineage tracing experiments and in long-term loss-of-function experiments, whole mesentery GFP+ areas in CD31+ SOX17+ areas were quantified using Photoshop and ImageJ softwares. For *Aplnr-CreERT2*-controlled GFP expression experiments, high magnification images close to the intestine were analyzed. For whole-mount intestinal tissue analysis, GFP+, SOX17+ and intestinal tissue area were quantified using Photoshop and ImageJ softwares. For Caspase-3 staining analysis in whole-mount intestinal tissue, Caspase-3 intensity in CD31+ and GFP+ areas were quantified in 3D using Imaris software. In the 10-h Esm1 lineage tracing experiment using E14.0 embryonic intestine, total ERG+, ERG+ Esm1+, ERG+ GFP+, and ERG+ Esm1+ GFP+ cells were quantified using Photoshop and ImageJ software. For large mesenteric artery and intestinal submucosal artery analysis, at least 3 high magnification images were taken per sample. CD31+ SOX17+ artery area, length, GFP+ area and nuclear density were quantified using Photoshop and ImageJ softwares. GFP+ cells with changes in morphology (elongated *vs* round) were counted using Photoshop and ImageJ softwares.

In the embryonic intestinal villus lineage tracing experiments, GFP+ area in CD31+ SOX17+ and CD31+ SOX17− areas were quantified using Photoshop and ImageJ softwares. For villus blood capillary length and width, images were directly quantified in 3D using Imaris software. For villus blood and lymphatic vessels, ERG+ nucleus, SOX17+ nucleus and GFP+ 3D volume and intensity analysis, images were quantified using Imaris software. In *Itgb1*[iEsm1KO], *Itgb1*[iEsm1HET] and control littermates, 6-13 villi per sample were analyzed. In *Vegfc*[iBmxKO] and control littermates, 12-29 villi per sample were analyzed. Villus SOX17+ arterial branch diameter and proportion of villus with more than one SOX17+ branch was quantified using Imaris, Photoshop and ImageJ softwares. In MAZ51-treated and DMSO control embryo experiments, 4-5 images comprising around 3 villi each were analyzed. Villus Caspase-3 intensity was directly quantified in 3D using Imaris software. In *Flt4*[iAplnrKO] and control littermates, 20-26 villi per sample were analyzed. In *Flt4*[iEsm1KO], *Flt4*[iEsm1HET] and control littermates, 16-31 villi per sample were analyzed. Villus VEGFR3 areas were quantified in 3D using Imaris software. For Esm1, LAMA4, COLIV, VEGFR3 and CD31 intensity quantifications, 7-15 villi per sample were analyzed. Each villus was divided into top and bottom villus capillary network area. In both top and bottom areas, vessel volume and staining intensities were quantified using Imaris software. For proliferation analyses, cells were double-stained for ERG and/or SOX17 and EdU. For embryonic intestinal villi, 6-9 high magnification images were analyzed. For embryonic mesenteries, 3 mesenteric artery branches per sample were imaged and analyzed. Double ERG and EdU masks were generated using Imaris software and cells were quantified using Photoshop and ImageJ softwares, counting as positive only cells in which all ERG+ area was EdU+.

In *Itgb1*[iEsm1HET] and *Itgb1*[iEsm1KO] postnatal experiments, whole retinas were analyzed using Photoshop and ImageJ softwares. For Esm1+ area quantification in adult villi, 5 images per animals, comprising around 3 villi each were quantified. CAV1+ Esm1+ and CAV1+ EMCN+ Esm1+ masks were generated using Imaris software and areas were quantified using Photoshop and ImageJ softwares. For GFP+ area quantification in EMCN+ and EMCN− of the adult villus, 9-12 images comprising around 3 villi each were analyzed. VEGFR2+ GFP+ and VEGFR2+ EMCN+ GFP+ masks were generated using Imaris software and areas were quantified using Photoshop and ImageJ softwares.

The number of embryos ($n = X$) analyzed is indicated for each image in the corresponding figure legend. Differences were considered statistically significant at $P < 0.05$. Data are shown as mean ± SD.

Statistical analyses were performed with Graphpad Prism9 v.9.4.0. For groups with equal variance, the unpaired t-test was used, whereas Welch's t-test was used for groups with unequal variance. Comparisons among multiple groups were done using either one-way ANOVA or Browne-Forsythe and Welch ANOVA with a specific post-hoc test for multiple comparisons. The source data used for quantitative analysis, as well as detailed statistical results are provided in the Source Data file.

### Reporting summary

Further information on research design is available in the Nature Portfolio Reporting Summary linked to this article.

## Data availability

The scRNA-seq data generated in this study have been deposited in the Gene Expression Omnibus (GEO): https://www.ncbi.nlm.nih.gov/geo/query/acc.cgi?&acc=GSE275961. Previously published scRNA-seq datasets analysed in conjunction with data from this study can be found at GSE175895 Zarkada et al., P6 WT. All measurements used for quantification are provided in the Source Data file along with the results of the different statistical analysis performed. Source data are provided in this paper. Source data are provided with this paper.

## Code availability

The custom code used for scRNA-seq analysis in this study is based on existing packages and own contributions. It is deposited in a publicly available database and can be accessed through the following link: https://doi.org/10.17617/3.Z8KMT8.

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

## Acknowledgements

We thank Dr. Bong-Ihn Koh for FACS analysis and sorting experiments. We thank Dr. Kishor K. Sivaraj for sharing *Hey1-eGFP* mice and Dr. Hongryeol Park for sharing *Aplnr-CreERT2* mice. For sharing RNAscope protocols, reagents and discussions, we would like to thank Dr. Clémentine Villeneuve and Claudia Ortmeier from the Wickström's department of the Max Planck Institute of Münster. We also thank Dr. Rodrigo Diéguez-Hurtado, Frank Berkenfeld and Silke Schröder for discussions and mouse colony maintenance. Animal, FACS, Genotyping, BioOptic and Bioinformatics facilities of the Max Planck Institute of Münster are gratefully acknowledged. The study was supported by the Max Planck Society (R.H.A.), the DFG Collaborative Research Center 1348 Project A10 (R.H.A., M.E.P., E.B.), the Leducq Foundation (R.H.A.) and the Cells in Motion (CiM) graduate school (V.M).

## Author contributions

E.B. and R.H.A. designed the study. E.B. performed the majority of the experiments. E.C.W. and E.B. performed the single-cell sequencing, and K.K. performed the bioinformatic transcriptomic analysis. V.M. generated samples for RNAscope. M.S. helped set up staining for FACS and perfomed FACS-sorting experiment. M.E.P. and F.B. generated P5 WT retina results and generated some of the P6 *Itgb1* mutant retina samples. M.L.K. provided critical mouse models and information for RNAscope experiments. E.B. and R.H.A. wrote the manuscript.

## Funding

## Competing interests

The authors declare no competing interests.
