## [Peer Review file · Nature Communications]

Artery formation in the intestinal wall and mesentery by intestine-derived Esm1+ endothelial cells

Corresponding Author: Professor Ralf Adams

Version 0:

Reviewer comments:

Reviewer #1

(Remarks to the Author)

Research on angiogenesis is actively being conducted, but how organ-specific vascular structures are formed remains unclear. The authors have analyzed the molecular mechanisms of vascular formation in the intestine and demonstrated the potential importance of ESM1-positive cells in arterial formation in the gut. The research group has previously achieved numerous outstanding results in vascular studies, including demonstrating that, in the retina, tip cells contribute to arterial formation by migrating retrogradely from the leading edge of blood vessels.

In this study, various genetic models, including ESM1-creERT2, were utilized to elucidate the fate and function of ESM1-positive cells in the intestine and their molecular mechanisms. Particularly noteworthy are the findings that integrin $\beta 1$ may be involved in the morphogenesis of ESM1-positive cells and arterial vessel diameter, and that VEGF-C and VEGFR3 may play roles in arterial formation. These are highly novel and intriguing results. Based on extensive data, this study represents an excellent contribution to clarifying the mechanisms of intestinal vascular formation.

On the other hand, the lineage tracing analyses, and certain gene expression analyses do not seem to provide sufficiently robust results. Below are some comments aimed at improving the manuscript.

1. Throughout the manuscript, the ROSA26-mTmG reporter mouse is used in several experiments. As I understand it, in this mouse model, cells are tomato-positive in the absence of Cre recombination and become GFP-positive upon recombination. Why is tomato positivity not shown across the entire tissue throughout the manuscript? If any specific modifications were made, an explanation is needed to ensure clarity.

2. As the name suggests, ESM1 is known to be EC-specific; however, in some images, including Fig. 1D, Fig. 7G, and Supplementary Fig. 1D, it appears to adhere around blood vessels, resembling pericytes. I think it is essential to verify whether ESM1-positive cells in the developing intestine express pericyte markers or not.

3. Regarding lineage tracing, in Fig. 1, Supplementary Fig. 5, and Supplementary Fig. 9, the authors analyze at consistent time points while varying the timing of 4OHT administration. While this approach has its merits, for lineage tracing, analyses using consistent administration timings should also be included. For example, administering 4OHT at E13.5 and observing the time course at 1 day (E14.5), 3 days (E16.5), and 5 days (E18.5) could help clarify the cell fate of GFP-positive cells. In the current model, there is a possibility that different cells are being labeled, which complicates the interpretation. Additionally, it is unclear how long Cre-mediated recombination continues to occur after one shot of 4OHT administration, making the interpretation of lineage tracing results challenging.

4. In the model using ESM1-CreERT2 where tamoxifen is administered at E13.5 and observed at E18.5 (Fig. 1D), why are there so few GFP-positive cells detected at E18.5? In Supplementary Fig. 1, immunostaining shows a substantial number of ESM1-positive cells in the intestine. However, in the lineage tracing model at E18.5, GFP-positive cells are only observed at mesentery and not observed in the intestine. If all the GFP-positive cells from E13.5 are migrating to the mesentery, more GFP-positive cells should be detectable. Are all ESM1-positive cells migrating? Is there any reduction in GFP-positive cells due to cell death? This point needs further investigation and clarification.

5. Regarding the results of the single-cell analysis, the cells are classified into arteries and veins. However, most of the blood vessels in the villi are capillaries. Why were capillary endothelial cells not detected? This point requires clarification.

6. In Fig. 4e, immunostaining shows that LAMA4, COLIV, and VEGFR3 are highly expressed in the distal parts of the villi. While the functional analysis of VEGFR3 is addressed in Fig. 8, what does it mean that "LAMA4 and COLIV decorate the capillaries in the distal villus"? Since these factors are expressed in more cells than ESM1, do ESM1-positive cells arise from vascular endothelial cells that express these factors? This point needs further clarification.

7. Regarding the inhibition of VEGFR3 by MAZ51, it is important to verify whether MAZ51 induces cell death. This should be addressed to ensure a comprehensive understanding of its effects.

8. Regarding the Emcn-negative, GFP-positive cells observed in Supplementary Fig. 1c, it is known that Emcn is expressed in vascular ECs and some hematopoietic cells from the early stages of development. Are these Emcn-negative cells truly vascular ECs? Or are they progenitor cells that only partially exhibit the characteristics of vascular ECs? This should be commented. In relation to this, the distinction between arteries and veins in villous blood vessels using EMCN and CAV1 staining in Supplementary Fig. 3 is a highly significant. Is this expression pattern specific to the villi of the intestine? Furthermore, do the single-cell analysis results show a similar result, with stronger EMCN expression in veins and stronger CAV1 expression in arteries?

9. It is very interesting to know whether the gene expression profiles of ESM1-positive vascular ECs in adults differ from those in the fetal stage. This distinction could provide valuable insights into the developmental roles and functional transitions of ESM1-positive cells.

Reviewer #2

(Remarks to the Author)

The manuscript by Bovay et al provides an extensive developmental analysis of intestinal endothelial (EC) progenitor cells that express Esm1, a classical tip cell marker. This overall rigorous work uses elegant genetic tools and some bioinformatics analysis to determine the fate and role of the Esm1 cells in the developing mouse intestine, and lineage tracing data supports contributions to both intestinal and mesenteric arteries from the villus and capillaries. Although contribution of Esm1+ EC to arteries is documented in the postnatal retina, this is the first description of their role in the intestine. A significant amount of data is presented without much explanation. Downstream Esm-Cre mediated deletion of integrin b1 reveals thinner arteries with more rounded EC, and genetic loss of VEGFR3 signaling (via *Aplnr-CreERT2*) affects artery diameter, suggesting that migration and/or morphology is important in the timeline. However, these manipulations do not appear to affect physiological function during late development. The significance of the study is that intestinal Esm1+ EC participate in vascular remodeling that is crucial for proper artery development in the intestine and mesentery. There are several suggestions to strengthen the manuscript.

Major issues:

1. The wealth of data is impressive; however, it is sometimes difficult to extract the key findings and potential significance. How the different timing of reporter excision leads to temporal conclusions regarding the Esm1+ cells is not clearly explained. The scRNA seq comparisons to retinal Esm1+ cells reveal significant overlap but some differences that are not well-explored. The cell cycle status of the Esm1+ cells and other populations is discussed, and this study found some evidence of proliferation in more mature arteries, but these findings were not explored further or their significance explained. The distal vs. proximal location of Esm1+ cells in the villi is documented and linked to VEGFR3 expression, but these spatial findings are not extended. How integrin b1 normally regulates Esm1+ EC function to provide proper arterialization is not expanded upon. Perhaps a visual model at the end would help coalesce the different data outputs.

2. Somewhat related to point 1, it is not clear how the intestinal Esm1+ EC differ from the retinal pool in ways that could affect their function or contributions differentially. Is b1 integrin important in the retinal Esm1+ cells? Here the signaling of VEGFC/VEGFR3 is shown to influence arterial diameter, but what is the role of VEGFA and does it differ from that of VEGFC? Why is the cell cycle status relevant and is that different from the retinal paradigm?

3. It would be interesting to comment more on the potential effects of reduced arterial diameter - would one predict that flow parameters might change and that could, over time, lead to pathology?

Minor issues:

1. Figure 1 is dense and difficult to navigate. In panel a there is no yellow box on left to show where the zoom on right is located. The timing is important to infer movement, but graphs are somewhat confusing - for example panel h is likely the vessels in the villus but vaguely labeled, and panel i is very difficult to distinguish the time points.

2. Figure 2c seems the same as Supp. Fig 4b - can this be condensed?

3. Figure 3b, the G2/M population in Venous EC1 and Venous EC2 seems very high compared to S phase values, please comment.

Version 1:

Reviewer comments:

Reviewer #1

(Remarks to the Author)

In the revised manuscript, Bovay et al. have addressed all of my concerns raised in the initial submission. They have conducted a substantial number of additional experiments and successfully resolved my concerns including the lineage tracing analyses and cell annotation and specificity of ESM1 cells. By employing multiple genetically modified models, multicolor immunostaining techniques, and scRNA-seq analysis, they have elucidated the developmental process of intestinal arteries.

Notably, their clarification of the mechanism by which Esm1⁺ progeny contribute to arterial endothelial cells represents a significant finding in vascular biology. In the Discussion, they touch upon the molecular similarities between the intestine, retina, and ischemic heart, and I believe that investigating Esm1-positive ECs in other organs is also a promising area of research.

From this reviewer, there are no further suggestions. This is a highly important study, and I strongly recommend it for publication.

REVIEWER COMMENTS

We would like to express our appreciation and gratitude to the reviewers for their constructive comments and insightful suggestions, which have greatly improved the quality of our manuscript. While a detailed point-by-point response to each comment is provided below, we would like to begin with a summary of the changes and additions made in the revised version of the paper.

Changes in the **Results** section:

- a) Exclusion of cell apoptosis in *Esm1*-derived GFP⁺ cells in E14.5 intestines and mesenteries, 24 hours post-treatment.
- b) Reorganization of the manuscript, with *Bmx*- and *Aplnr-CreERT2*-controlled lineage tracing experiments now presented at the beginning of the manuscript.
- c) Additional analysis of early lineage tracing experiments initiated at E13.5 and analyzed at E14.5.
- d) Supplementary analysis of pooled blood endothelial cell (EC) scRNA-seq results, including the identification of subclusters composed of capillary ECs from the intestine.
- e) New postnatal retina results showing the effects of *Itgb1* deletion in *Esm1*-derived cells, with comparison to the embryonic phenotype and additional interpretation of these results.
- f) Exclusion of cell apoptosis in embryonic intestines treated with MAZ51.

Changes in the **Methods** section:

- a) Inclusion of treatment details used for postnatal *Itgb1* deletion in *Esm1* cells.
- b) scRNA-seq preprocessing includes now the tricycle¹ method used to predict cell-cycle phase.
- c) Inclusion of detailed information regarding quantitation of new lineage tracing experiments and lineage tracing experiments with cleaved Caspase-3 or *Esm1* staining.
- d) Update of Table 1 with cleaved Caspase-3 antibody information included.

Changes in the **Main Figures**:

In general, since the majority of blood vessels in embryonic villi are composed of capillary ECs, the pooled blood EC scRNA-seq UMAP plots now show "venous" and "arterial" regions, which includes peri-arterial and peri-arterial capillaries, respectively.

Fig. 1a: A yellow box has been added to highlight the region shown in the high-magnification image.

Fig. 1f: Graph labels were modified to improve clarity.

Fig. 1h, i: To enhance readability, the graph showing the percentage of GFP⁺ area found in SOX17⁺ versus SOX17⁻ regions of the villus vasculature now displays only the E17.5 time point. Graph labels have also been revised for clarity.

Fig. 3a: An updated UMAP plot was generated using the tricycle method¹.

Fig. 3b: To avoid confusion, the graph showing the proportion of cells in G2/M, S, or G1 phase in each EC subgroup after cell cycle regression has been moved to Supplementary Fig. 9c.

Fig. 5h: A model illustrating mesenteric artery development following long-term *Itgb1* deletion in *Esm1* cells has been added.

Changes in the **Supplementary Figures**:

Supplementary Fig. 1d: A single focal plane now shows the area outlined in the left panel.

Supplementary Fig. 1e: High-magnification images and single focal planes have been added to identify sprouting and capillary mesenteric ECs.

Supplementary Fig. 2: New *Esm1* lineage tracing results at E14.0/E14.5 have been added to rule out *Esm1*⁺ cell apoptosis and to assess recombination efficiency.

Supplementary Fig. 3: Former Supplementary Fig. 9a-d.

Supplementary Fig. 4: Former Supplementary Fig. 5. The right panel of Supplementary Fig. 4i was replaced to show the mesenteric vasculature close to the intestine, similar to other time points and representative of the quantifications. Additionally, arrows were added to improve clarity.

Supplementary Fig. 5: New results from *Esm1*⁻, *Bmx*⁻, and *Aplnr*-CreERT2-controlled lineage tracing experiments initiated at E13.5 are presented. A schematic representation explaining each model has also been included in Supplementary Fig. 5g.

Supplementary Fig. 6: Former Supplementary Fig. 2.

Supplementary Fig. 7: Former Supplementary Fig. 3. Graph labels have been improved for better clarity and transparency.

Supplementary Fig. 8: Former Supplementary Fig. 4. New UMAP plots showing expression levels of venous, capillary and arterial EC genes have been added in Supplementary Fig. 8c.

Supplementary Fig. 9b, c: Former Supplementary Fig. 6b. UMAP and bar plots showing cells in G2/M, S or G1 phase after cell cycle regression have been updated using the tricycle method¹. The bar plot now includes the total number of cells in each EC subgroup, along with the proportion in each cell cycle phase.

Supplementary Fig. 10b: Former Supplementary Fig. 7b. The volcano plot was updated to include more relevant genes, such as *Itgb1*.

Supplementary Fig. 11a: Former Supplementary Fig. 8a. Relevant biological processes and cellular components are now highlighted in red.

Supplementary Fig. 11b: Former Supplementary Fig. 8b. High-magnification images of the villi are added.

Supplementary Fig. 12: New results on *Itgb1* deletion in Esm1⁺ cells in the postnatal retina are now included.

Supplementary Fig. 13: Former Supplementary Fig. 9e-h.

Supplementary Fig. 14: New data show minimal activated Caspase-3 staining in E18.0 embryonic intestines following MAZ51 or control treatment.

We are confident that the new results, along with the thorough revision of the text and improvements to the figures, have substantially enhanced the quality and readability of our study.

In the following point-by-point response, each reviewers' comments is shown in full and followed by our reply.

Reviewer #1 (Remarks to the Author):

Research on angiogenesis is actively being conducted, but how organ-specific vascular structures are formed remains unclear. The authors have analyzed the molecular mechanisms of vascular formation in the intestine and demonstrated the potential importance of ESM1-positive cells in arterial formation in the gut. The research group has previously achieved

numerous outstanding results in vascular studies, including demonstrating that, in the retina, tip cells contribute to arterial formation by migrating retrogradely from the leading edge of blood vessels.

In this study, various genetic models, including ESM1-creERT2, were utilized to elucidate the fate and function of ESM1-positive cells in the intestine and their molecular mechanisms. Particularly noteworthy are the findings that integrin $\beta 1$ may be involved in the morphogenesis of ESM1-positive cells and arterial vessel diameter, and that VEGF-C and VEGFR3 may play roles in arterial formation. These are highly novel and intriguing results. Based on extensive data, this study represents an excellent contribution to clarifying the mechanisms of intestinal vascular formation.

On the other hand, the lineage tracing analyses, and certain gene expression analyses do not seem to provide sufficiently robust results. Below are some comments aimed at improving the manuscript.

1. Throughout the manuscript, the ROSA26-mTmG reporter mouse is used in several experiments. As I understand it, in this mouse model, cells are tomato-positive in the absence of Cre recombination and become GFP-positive upon recombination. Why is tomato positivity not shown across the entire tissue throughout the manuscript? If any specific modifications were made, an explanation is needed to ensure clarity.

We thank the reviewer for highlighting this aspect of the ROSA26-mTmG reporter mouse line. All cells that have not been exposed to Cre-mediated recombination express membrane-anchored tdTomato with varying intensities depending on the cell type, resulting in widespread red fluorescence throughout all developmental stages². We therefore reasoned that the red signal is negligible for the purpose of study and showed only the green GFP signal in our manuscript. Nonetheless, we are providing here for the reviewers some examples of tdTomato fluorescence in the embryonic intestine and mesentery (Reviewer1 Fig. 1a). These experiments, which were used to trace the progeny of Esm1⁺, Bmx⁺, and Aplr⁺ cells, also address Reviewer 1, Question 3 (Reviewer1 Fig. 3). Broad tdTomato fluorescence can be seen throughout the entire tissue but is excluded from areas with robust Cre recombination, indicated by GFP expression.

Additionally, we include data from adult intestines, where the tdTomato signal is also found throughout the whole tissue (Reviewer1 Fig. 1b; from the same experiment shown in

Supplementary Fig. 7g of the revised manuscript). These results demonstrate that the tdTomato signal shows the expected broad expression throughout the tissue.

2. As the name suggests, ESM1 is known to be EC-specific; however, in some images, including Fig. 1D, Fig. 7G, and Supplementary Fig. 1D, it appears to adhere around blood vessels, resembling pericytes. I think it is essential to verify whether ESM1-positive cells in the developing intestine express pericyte markers or not.

We thank the reviewer for raising this very important point. In our scRNA-seq experiments, *Esm1* expression is exclusively detected in the blood vascular EC population (*Cdh5*⁺ *Prox1*⁻; see Reviewer1 Fig. 2a). We have also included known pericyte markers (*Pdgfrb* and *Cspg4*)³, which are expressed in our scRNA-seq experiments in the cell group labeled “Pericytes and vascular smooth muscle cells (vSMC)”. Pericytes express both *Pdgfrb* and *Cspg4* but are negative for *Cdh5* (pan-endothelial marker), *Prox1* (lymphatic marker) and *Esm1*.

We also include high magnification images from Fig. 1d of the manuscript, showing GFP⁺ sprouting and capillary cells surrounding the large mesenteric artery (Reviewer1 Fig. 2b). Single focal plane images clearly confirm that GFP⁺ cells co-localize with CD31 staining. These results further validate *Esm1* expression in the sprouting cells that form the mesenteric capillary network, as indicated in Fig. 1a. Additionally, we provide high magnification single focal plane images of results from Fig. 7g, showing *Esm1*-derived GFP⁺ cells (Reviewer1 Fig. 2c). These images confirm the contribution of GFP⁺ cells to the submucosal vascular network specifically. All these results indicate specific *Esm1* expression and contribution to the blood vascular ECs of the embryonic intestine and mesentery.

Finally, in the E12.5 mesentery, *Aplnr*⁺ cells express GFP in SOX17⁻ CD31⁺ ECs 24 hours post-induction (Reviewer1 Fig. 2d, from Supplementary Fig. 1d), indicating that *Aplnr* is also a blood vascular EC-specific marker.

Some of these results are now included in the revised manuscript in Supplementary Fig. 1d and Supplementary Fig. 1e.

3. Regarding lineage tracing, in Fig. 1, Supplementary Fig. 5, and Supplementary Fig. 9, the authors analyze at consistent time points while varying the timing of 4OHT administration. While this approach has its merits, for lineage tracing, analyses using consistent administration timings should also be included. For example, administering 4OHT at E13.5 and observing the

time course at 1 day (E14.5), 3 days (E16.5), and 5 days (E18.5) could help clarify the cell fate of GFP-positive cells

In the current model, there is a possibility that different cells are being labeled, which complicates the interpretation. Additionally, it is unclear how long Cre-mediated recombination continues to occur after one shot of 4OHT administration, making the interpretation of lineage tracing results challenging.

We thank the reviewer for raising this important point. As already described in the mouse embryo, the kinetics following 4-OHT or tamoxifen administrations vary significantly. 4-OHT peaks in the mouse serum after around 6 hours after administration and declines relatively rapidly in the following 24 hours post-induction^{4,5}. Tamoxifen undergoes conversion into 4-OHT and other metabolites in the liver, which takes around 6–12 hours *in vivo*. Its levels peak at 12 hours post-induction and can remain up until 72 hours^{4,5}. For these reasons, we specifically performed lineage tracing experiments with a single dose of 4-OHT throughout our study.

We now also include additional lineage tracing experiments initiated at E13.5 and analyzed 24 hours post-induction, to show acute recombination induced by *Esm1*, *Bmx* and *Aplnr*-controlled CreERT2 expression in the E14.5 embryonic intestine and mesentery as a starting point (Reviewer1 Fig. 3). Combining these new lineage tracing experiments with experiments already described in Fig. 1, Supplementary Fig. 5, and Supplementary Fig. 9, allow the reader to follow the fate of recombined cells from E13.5 to E18.0/E18.5. These new results, which are now included as Supplementary Fig. 5 of the revised manuscript, confirm the conclusions of our previous lineage tracing experiments. Most importantly, they clearly show that *Esm1*⁺ cells are initially labelled in the intestine but show up later in arteries of the intestinal mucosa and mesentery.

We also include a scheme summarizing the findings derived from the lineage tracing experiments (Reviewer1 Fig. 3g).

4. In the model using ESM1-CreERT2 where tamoxifen is administered at E13.5 and observed at E18.5 (Fig. 1D), why are there so few GFP-positive cells detected at E18.5? In Supplementary Fig. 1, immunostaining shows a substantial number of ESM1-positive cells in the intestine. However, in the lineage tracing model at E18.5, GFP-positive cells are only observed at mesentery and not observed in the intestine. If all the GFP-positive cells from E13.5 are migrating to the mesentery, more GFP-positive cells should be detectable. Are all

ESM1-positive cells migrating? Is there any reduction in GFP-positive cells due to cell death? This point needs further investigation and clarification.

We thank the reviewer for raising this important point. Several factors limit the number of GFP⁺ cells contributing to the mesenteric arteries. Esm1 expression is transient, and since we perform only one 4-OHT injection, we label cells only within the 24-hour time window following the injection, excluding any Esm1⁺ cells generated before or after the short period of Cre-mediated recombination⁴. To address the possibility of losing cells, we now provide evidence that GFP⁺ cells, 24 hours after lineage tracing from E13.5, do not co-stain with the apoptotic marker activated caspase-3 (Reviewer1 Fig. 4a, b). Although some Caspase3⁺ ECs were occasionally detected, no increase in GFP⁺ cells was observed compared to the CD31⁺ vascular network (Reviewer1 Fig. 4a, b). It is also important to note that some GFP⁺ cells had already migrated outside the intestinal mucosa toward the mesenteric vascular network, as shown in Reviewer1 Fig. 3a.

As discussed above, serum 4-OHT peaks in the mouse around 6 hours post-injection. We then tested the proportion of GFP⁺ cells obtained at E13.5 after only 10 hours post-induction. After 10 hours, GFP⁺ cells were already detectable in the intestine (Reviewer1 Fig. 4c-e). However, co-staining with Esm1 shows that only around 20% of Esm1⁺ cells become GFP⁺ (Reviewer1 Fig. 4d, e). These results suggest that Esm1-derived cells are not lost, but that we rather only label cells expressing the highest levels of Esm1 at the time of injection, which likely underestimates their contribution to the arterial network. These new results are now included in the revised manuscript in Supplementary Fig. 2.

5. Regarding the results of the single-cell analysis, the cells are classified into arteries and veins. However, most of the blood vessels in the villi are capillaries. Why were capillary endothelial cells not detected? This point requires clarification.

We thank the reviewer for drawing our attention to this aspect of the data presentation. It is, of course, correct that the intestinal vasculature contains substantial amounts of capillary ECs, which is supported both by our immunostaining and scRNA-seq results. It is also known that blood vessels are zoned along the arteriovenous axis, generating a seamless continuum of ECs with gradual transcriptomic changes, as was nicely established by the group of Christer Betsholtz⁶. Unsupervised clustering of the cells in our dataset also revealed gradual vascular zonation throughout the entire vascular tree, with an overlap of early arterial and venous

markers in the *Esm1*⁺ subcluster (Reviewer1 Fig. 5a, from manuscript Fig. 2c). We now also include capillary markers (*Pcdh12*, *Gpihbp1* and *Prdm1*; Reviewer1 Fig. 5b)^{7,8}. These results show that the *Esm1*⁺ group consists of capillary ECs but also that parts of the Venous EC2, Arterial EC1, and Arterial EC2 groups express capillary EC markers. In contrast, the Venous EC1 and Arterial EC3 clusters represent differentiated (and more mature) venous and arterial ECs, respectively. We have now added these results to Supplementary Fig. 8 and made corresponding modifications in the revised manuscript.

6. In Fig. 4e, immunostaining shows that LAMA4, COLIV, and VEGFR3 are highly expressed in the distal parts of the villi. While the functional analysis of VEGFR3 is addressed in Fig. 8, what does it mean that "LAMA4 and COLIV decorate the capillaries in the distal villus"? Since these factors are expressed in more cells than ESM1, do ESM1-positive cells arise from vascular endothelial cells that express these factors? This point needs further clarification.

We thank the reviewer for this important question. The markers LAMA4, COLIV, and VEGFR3 are known to be expressed by retinal tip cells^{9,10}. Similar to the retina, we observed that VEGFR3, COLIV, and LAMA4 staining is highest in the distal villus, the region containing *Esm1*⁺ cells (Fig. 4e-g). However, the UMAP plots in Fig. 4d also show *Col4a2* and *Lama4* transcripts in other EC subclusters and in venous ECs. It is indeed very likely that lineage tracing of venous ECs would reveal contribution to all other EC populations, as has been previously shown for the retina¹¹.

7. Regarding the inhibition of VEGFR3 by MAZ51, it is important to verify whether MAZ51 induces cell death. This should be addressed to ensure a comprehensive understanding of its effects.

We thank the reviewer for raising this concern. We tested for activated Caspase3 staining in both DMSO- and MAZ51-treated samples (Reviewer1 Fig. 6a, b). Some Caspase3⁺ cells were observed in the intestinal tissue (Reviewer1 Fig. 6a). However, no increase in Caspase3⁺ cells was observed in the intestinal vasculature, nor in GFP⁺ cells following MAZ51 treatment (Reviewer1 Fig. 6b), suggesting that the treatment does not induce endothelial apoptosis. These new results are now included in the revised manuscript in Supplementary Fig. 14.

8. Regarding the *Emcn*-negative, GFP-positive cells observed in Supplementary Fig. 1c, it is

known that Emcn is expressed in vascular ECs and some hematopoietic cells from the early stages of development. Are these Emcn-negative cells truly vascular ECs? Or are they progenitor cells that only partially exhibit the characteristics of vascular ECs? This should be commented.

Thank you for this comment. We now provide results showing Bmx⁺ cell-derived, genetically labeled progeny at E14.5, 24 hours after lineage tracing from E13.5 (Reviewer1 Fig. 7). High magnification images show that labeled cells are located within the CD31⁺ SOX17⁺ arterial vasculature, consistent with what has been described previously¹². However, consistent with a role of Bmx in the growth and differentiation of hematopoietic cells¹³, we occasionally observed GFP signal in round CD31⁻ SOX17⁻ cells outside of the vasculature (indicated by asterisks in Reviewer1 Fig. 7). Nevertheless, Bmx⁺ cells within the mesenteric and intestinal CD31⁺ SOX17⁺ vasculature are clearly arterial ECs.

9. In relation to this, the distinction between arteries and veins in villous blood vessels using EMCN and CAV1 staining in Supplementary Fig. 3 is a highly significant. Is this expression pattern specific to the villi of the intestine? Furthermore, do the single-cell analysis results show a similar result, with stronger EMCN expression in veins and stronger CAV1 expression in arteries?

We thank the reviewer for raising these important points. In the adult mouse intestine, Endomucin (EMCN) serves as a venous marker and has previously been shown to be expressed in veins and venules of the intestinal villus^{14,15}. In this study, we refer again to the results presented in Supplementary Fig. 3 (new version Supplementary Fig. 7a), which clearly demonstrate strong EMCN staining in the veins and venules (Reviewer1 Fig. 8a). In contrast, Caveolin-1 (CAV1) is broadly expressed across the entire vascular network of the adult villus. We now include the single staining for CAV1, demonstrating that its expression in the embryonic intestine is not restricted to arteries but also extends to villus venules and veins (Reviewer1 Fig. 8b). In the same sample, EMCN staining is broadly distributed throughout the entire villus vascular network and co-localizes with CAV1, suggesting that the embryonic vasculature is still undergoing development and maturation (Reviewer 1 Fig. 8b). Notably, (CAV1⁺ SOX17⁺) arteries in the submucosa beneath the villi lack EMCN expression. These findings are further supported by 24-hour Bmx lineage tracing experiments (Supplementary Fig. 3d of the revised manuscript) and our scRNA-seq data (Reviewer1 Fig. 8c), indicating

upregulation of *Bmx* and downregulation of *Emcn* in the Arterial EC3 population, which represents the endothelium of the most mature arteries (Reviewer1 Fig. 8c).

10. It is very interesting to know whether the gene expression profiles of ESM1-positive vascular ECs in adults differ from those in the fetal stage. This distinction could provide valuable insights into the developmental roles and functional transitions of ESM1-positive cells.

As previously described, *Esm1* is expressed in the villus capillaries, particularly at the villus apex, where high VEGF signaling promotes the reorganization of EC junctions to enhance nutrient uptake in the adult mouse intestine^{16,17}. In the current study (Supplementary Fig. 7 of the revised manuscript), we also show that *Esm1* is specifically enriched in the EMCN-negative peri-arterial side of the adult villus capillary network. To further validate these findings and compare gene expression profiles between embryonic and adult stages, we integrated our data with three published adult datasets from Kalucka et al., Niec et al., and Wiggins et al.^{18–20}. We subclustered the various intestinal blood EC populations, focusing on those expressing *Esm1*. These included the *Esm1* population in the embryonic sample and 3 populations in the combined adult samples that showed high expression of capillary markers such as *Prdm1*, *Pcdh12*, and *Rgcc* (Reviewer1 Fig. 9a–c). Additionally, a fourth capillary population (*Aqp7*⁺), already identified in the adult villus capillary network^{19,20}, shows some *Esm1* expression (Reviewer1 Fig. 9a–c).

As expected, we observe strong expression of *Bmx*, *Gja4*, and *Hey1* in arterial EC populations, whereas venous ECs are positive for *Nr2f2*, *Madcam1* and *Fam174b* (Reviewer1 Fig. 9d). Interestingly, the capillary population “*Esm1*⁺ EC1” in the adult dataset shows low expression of arterial markers (*Gja4* and *Hey1*) but is positive for the venous marker *Fam174b*. In contrast, the two other major capillary populations, “*Esm1*⁺ EC2” and “*Esm1*⁺ EC3,” exhibit high expression of arterial markers. These findings confirm that *Esm1* is expressed across a substantial portion of the heterogeneous adult villus capillary network, with predominant expression in peri-arterial capillaries within the intestinal villus.

Finally, analysis of the top markers expressed in embryonic and adult *Esm1*⁺ populations revealed substantial differences in gene expression across the four identified clusters (embryonic *Esm1*⁺ EC and adult *Esm1*⁺ EC1, EC2, and EC3). These may well reflect developmental stage-specific programs (as indicated by the expression of *Mest* and *Jund* in Reviewer1 Fig. 9e, f). Notably, certain genes are specifically upregulated in the embryonic

Esm1⁺ population, such as *Nid2* and *Adam19*, while others, including *Car4* and *Ramp3*, are confined to the adult populations (Reviewer1 Fig. 9f). Collectively, these results indicate that the villus capillary network undergoes significant transcriptional changes during development, reflecting functional specialization and maturation of the villus capillary network in the adult intestine.

Reviewer #2 (Remarks to the Author):

The manuscript by Bovay et al provides an extensive developmental analysis of intestinal endothelial (EC) progenitor cells that express *Esm1*, a classical tip cell marker. This overall rigorous work uses elegant genetic tools and some bioinformatics analysis to determine the fate and role of the *Esm1* cells in the developing mouse intestine, and lineage tracing data supports contributions to both intestinal and mesenteric arteries from the villus and capillaries. Although contribution of *Esm1*⁺ EC to arteries is documented in the postnatal retina, this is the first description of their role in the intestine. A significant amount of data is presented without much explanation. Downstream *Esm*-Cre mediated deletion of integrin $\beta 1$ reveals thinner arteries with more rounded EC, and genetic loss of VEGFR3 signaling (via *Aplnr*-CreERT2) affects artery diameter, suggesting that migration and/or morphology is important in the timeline. However, these manipulations do not appear to affect physiological function during late development. The significance of the study is that intestinal *Esm1*⁺ EC participate in vascular remodeling that is crucial for proper artery development in the intestine and mesentery. There are several suggestions to strengthen the manuscript.

Major issues:

1. The wealth of data is impressive; however, it is sometimes difficult to extract the key findings and potential significance. How the different timing of reporter excision leads to temporal conclusions regarding the *Esm1*⁺ cells is not clearly explained.

Thank you very much for your feedback. We trust that data presentation is improved in the revised manuscript. We now also provide short-term lineage tracing experiments (Reviewer1

Fig. 3; Reviewer1 Fig. 4), showing *Esm1*⁺ cell-derived progeny at early time points as well as the specific localization of *Esm1*⁺ cells stained for *Esm1* (Reviewer1 Fig. 4c). Additionally, we have included a schematic summary illustrating the temporal contribution of fate-tracked cells within the intestinal and mesenteric vasculature (Reviewer1 Fig. 3g). These new findings are incorporated into the revised manuscript as Supplementary Fig. 2 and Supplementary Fig. 5.

The scRNA seq comparisons to retinal *Esm1*⁺ cells reveal significant overlap but some differences that are not well-explored.

Thank you very much for this valuable comment, which has led us to explore this important aspect in the revised manuscript.

In the study by Zarakada et al.⁹, P6 tip cells were shown to upregulate markers such as *Kcne3*, *Sl00a6* and *Angpt2*, which are expressed at significantly lower levels in *Esm1*⁺ villus ECs (Reviewer2 Fig. 1a, b). In contrast, genes associated with vascular permeability and metabolism, such as *Plvap*, *Plpp3* and *Fabp4*, show higher expression in the embryonic intestine. Nonetheless, it is accurate that several retinal tip cell markers, including *Esm1*, *Flt4*, *Nid2*, *Lamb1*, *Col4a2* and *Itgb1*, are also expressed by *Esm1*⁺ villus ECs (Reviewer2 Fig. 1b). This led us to investigate whether the inactivation of *Itgb1*, which impairs *Esm1*⁺ cell migration, would differentially affect vascular development in the two organ systems despite the presence of a shared “core” gene signature.

As previously shown for pan-endothelial Cre-mediated deletion²¹ of *Itgb1*, inactivation of the gene in retinal *Esm1*⁺ cells results in defective sprouting and cell extension, leading to blunted regions at the leading edge of the growing vascular plexus (Reviewer2 Fig. 1c). This defect impairs overall vascular growth in the retina (Reviewer2 Fig. 1d). In contrast, deletion of *Itgb1* in intestinal *Esm1*⁺ cells does not compromise intestinal vascular development. High-magnification images clearly show the presence of GFP⁺ (recombined) ECs within the vascular network without clear evidence of impaired angiogenesis (Reviewer2 Fig. 1e).

However, we identified a critical role for *Itgb1* in the large mesenteric arteries, where it is essential for maintaining the stability of *Esm1*⁺ cell progeny within these vessels (revised manuscript Fig. 5). These findings highlight that, despite similar gene expression profiles, *Itgb1* inactivation in *Esm1*⁺ cells leads to divergent outcomes depending on the tissue context. These differences are most likely driven by distinct cellular mechanisms and microenvironments, resulting in tissue-specific phenotypic consequences. In the retina, *Esm1*-derived cells support sprouting angiogenesis, whereas in the embryonic gut, newly generated *Esm1*⁺ cells contribute

to the formation of large mesenteric arteries and require the integrin subunit to maintain stable arterial structures (revised manuscript Fig. 5). These new results are now incorporated into the revised manuscript in Supplementary Fig. 10, Supplementary Fig. 11 and Supplementary Fig. 12.

The cell cycle status of the Esm1⁺ cells and other populations is discussed, and this study found some evidence of proliferation in more mature arteries, but these findings were not explored further or their significance explained.

Thank you for this comment. Reduced cell proliferation is a well-established feature of Esm1⁺ and arterial ECs²²⁻²⁷, which prompted us to investigate this aspect in embryonic intestinal Esm1⁺ ECs (Fig. 3). In our scRNA-seq dataset, intestinal Esm1⁺ ECs show the expected low expression of cell cycle regulators, whereas these genes are again upregulated in the most mature arterial ECs, which we have directly confirmed by EdU labeling of mesenteric arteries (Fig. 3). We now provide additional supplementary data for the reviewer that highlights the relevance of cell proliferation within the mesenteric arterial network (Reviewer2 Fig. 2).

We went back to the comparison of our scRNA-seq dataset with P6 retinal ECs from Zarakada et al. (Reviewer2 Fig. 2a). Both retinal and embryonic intestinal Esm1⁺ ECs but also Bmx⁺ arterial ECs in the retina exhibit low proliferative activity (Reviewer2 Fig. 2a, b)²³⁻²⁷, whereas markers of cell proliferation are upregulated in the arterial EC3 subpopulation from embryonic intestine and mesentery (Reviewer2 Fig. 2b). This finding raised the possibility that arterial EC proliferation might be a compensatory mechanism providing developmental resilience when the system is challenged. Indeed, when arterial diameter in the mesentery is reduced after inactivation of *Itgb1* in Esm1⁺ cells (Fig. 5), SOX17⁺ EdU⁺ proliferating arterial ECs are increased relative to control littermates (Reviewer2 Fig. 2c, d). These findings indicate that mesenteric arterial ECs retain the capacity to proliferate, which might help to compensate for insufficient contribution of Esm1⁺ cells during arterial growth.

The distal vs. proximal location of Esm1⁺ cells in the villi is documented and linked to VEGFR3 expression, but these spatial findings are not extended.

It is indeed interesting that both Esm1 and Flt4/VEGFR3 are expressed in the upper region of the villus capillary network (Fig. 4), which reflects that both are VEGF-A-induced genes²⁸⁻³¹ and previous work has already established that the distal villus is a site of high VEGF-A

signaling¹⁶. In addition, we show that components of the basement membrane, such as Collagen IV and Laminin alpha4 (Fig. 4), are enriched around distal vessels and might influence cell signaling, specification and migration processes.

Pdgfra^{high} villus tip telocytes, previously identified in both the embryo and adult intestine^{16,32–35}, also exhibit a distinct top-to-bottom spatial distribution within the villus (Reviewer2 Fig. 3). All previous and new findings together indicate the presence of a specialized microenvironment at the villus tip, which is likely to direct vascular development and arterialization.

e. How integrin b1 normally regulates *Esm1*⁺ EC function to provide proper arterialization is not expanded upon. Perhaps a visual model at the end would help coalesce the different data outputs.

Thank you very much for this comment. As suggested, we have added a model showing the effect of *Itgb1* inactivation in the progeny of *Esm1*⁺ cells in the mesenteric artery (Reviewer2 Fig. 4). This scheme was included in the new manuscript in Fig. 5h. In our view, the conceptually most important finding here is that a genetic alteration of *Esm1*⁺ cells in the villus vasculature will compromise the behavior of these cells inside the mesenteric artery, which strongly supports the finding that villus-derived ECs give rise to arterial endothelium in the intestinal wall and mesentery.

2. a. Somewhat related to point 1, it is not clear how the intestinal *Esm1*⁺ EC differ from the retinal pool in ways that could affect their function or contributions differentially. Is b1 integrin important in the retinal *Esm1*⁺ cells?

Thank you very much for this question. We trust that our earlier answer has already addressed both the differences and common features of *Esm1*⁺ ECs in the retina and intestine. We trust that the data provided in Reviewer2 Fig. 1 clearly shows that the inactivation of *Itgb1* in the *Esm1*⁺ population leads to distinct phenotypes in the two tissues.

b. Here the signaling of VEGFC/VEGFR3 is shown to influence arterial diameter, but what is the role of VEGFA and does it differ from that of VEGFC?

Consistent with its well-established role as a master regulator of blood vessel growth, it is very likely that VEGF-A signaling is indispensable for all aspects of vascular growth in the intestine, as has been previously shown for the zebrafish embryo³⁶. This, however, also implies that it will be challenging to dissect the roles of VEGF-A in angiogenesis and artery formation. In contrast, much less is known about the function of VEGF-C and VEGFR3 in growing blood vessels so that we focused on these molecules rather than on VEGF-A and VEGFR2.

This decision was also supported by the analysis of our scRNA-seq dataset (Reviewer2 Fig. 5), which showed broad expression of *Kdr* (encoding VEGFR2) and *Flt1* (VEGFR1) in all EC subpopulations, which is further supported by immunostaining showing VEGFR2 both in the villus tip and base (Reviewer2 Fig. 5a–d). In contrast, *Flt4* (VEGFR3) is more restricted to the villus capillary and *Esm1*⁺ ECs in our scRNA-seq data and, by immunostaining, the receptor is confined to the upper region of the villus capillary network (Reviewer2 Fig. 5d, revised manuscript Fig. 4g, Fig. 8b and Supplementary Fig 15b). We also looked at the co-receptors *Nrp1* and *Nrp2* and found them enriched in the arterial/per-arterial and venous/peri-venous EC populations, respectively (Reviewer2 Fig. 5d).

Given our specific focus on *Esm1*⁺ cells, VEGFR3 clearly appeared as the most promising candidate regulator within the VEGF receptor family. Moreover, in contrast to *Vegfc*, which is predominantly expressed by arterial ECs, *Vegfa* is broadly expressed across multiple intestinal cell types, including ECs, epithelial cells and mesenchymal cells (Reviewer2 Fig. 5e). All these aspects together directed our investigation towards VEGF-C and VEGFR3.

c. Why is the cell cycle status relevant and is that different from the retinal paradigm?

Thank you for this question. Please refer to our answer provided earlier and the data shown in Reviewer2 Fig. 2.

3. It would be interesting to comment more on the potential effects of reduced arterial diameter - would one predict that flow parameters might change and that could, over time, lead to pathology?

Thank you for this important comment. Indeed, reduced arterial diameter has significant implications for critical blood flow parameters, impacting cardiovascular function and playing a key role in the etiology of many vascular diseases³⁷. As arteries narrow, blood flow decreases, whereas velocity and shear stress increase. Furthermore, post-stenotic regions are more likely

to exhibit turbulent flow, which is known to induce pro-inflammatory signaling in ECs and is associated with an increased risk of pathologies such as atherosclerosis³⁷.

However, it remains to be explored whether subsequent compensatory processes might lead to the restoration of normal arterial caliber. Increased proliferation of differentiated arterial ECs within the mesenteric artery (Reviewer2 Fig. 2c) could be such a compensatory response. Moreover, pruning and remodeling processes³⁸ as well as increased incorporation of Esm1⁺ cell progeny after the cessation of tamoxifen-induced Cre activity might also lead to the restoration of normal vessel calibers and perfusion. Future work will have to address this important and interesting question, as the exposure of pregnant dams to tamoxifen interferes with the normal delivery of the pups³⁹, precluding an easy extension of our current study to postnatal development.

Minor issues:

1. Figure 1 is dense and difficult to navigate. In panel a there is no yellow box on left to show where the zoom on right is located. The timing is important to infer movement, but graphs are somewhat confusing - for example panel h is likely the vessels in the villus but vaguely labeled, and panel i is very difficult to distinguish the time points.

We thank the reviewer for the feedback. We have revised the graphs, microscopic images and their labels (Fig. 1f, h, i). We trust that these modifications have improved the data presentation.

2. Figure 2c seems the same as Supp. Fig 4b - can this be condensed?

Thank you for the opportunity to clarify this point. Figure 2c displays a UMAP plot in which our three individual scRNA-seq datasets have been integrated. The individual datasets originate from separate sequencing experiments that are based on different tissue sources (mesentery vs. intestine) and approaches (single cell suspension from whole tissue vs. FACS-sorted ECs from intestine). UMAP plots for these three separate experiments are shown in Supplementary Fig. 8b of the revised manuscript (which was Supplementary Fig. 4b in the original submission). While much of our work has been guided by the fully integrated data, presentation of the individual scRNA-seq dataset adds further valuable information: a) it makes clear that the majority of ECs was recovered from the experiment involving FACS, b) it shows that the mature clusters Venous EC1 and Arterial EC3 are enriched in the mesentery, and c) it demonstrates that we obtained similar endothelial subclusters from the three different

approaches, arguing against potential experimental artefacts. Taken together, there are good reasons to present both the integrated data and the three individual scRNA-seq datasets separately, which also provides additional transparency regarding our data.

3. Figure 3b, the G2/M population in Venous EC1 and Venous EC2 seems very high compared to S phase values, please comment.

It is true that the G2/M population appeared higher than the S-phase population in the initial analysis. This discrepancy can be attributed to several factors. Primarily, cell cycle phase assignment relies on predefined static gene sets, which may not fully capture the complexity of in vivo conditions across different tissues. Additionally, absolute cutoffs are difficult to balance between G2/M and S-phase. In the revised manuscript, we now provide a clearer representation of cells in G2/M and S phases both before (Fig. 3a) and after cell cycle regression (Supplementary Fig. 9a–c) using the tricycle method¹. The updated analysis shows a more balanced distribution of G2/M and S-phase cells across the different EC subclusters (Supplementary Fig. 9c).

References

1. Zheng, S. C. *et al.* Universal prediction of cell-cycle position using transfer learning. *Genome Biology* **23**, 41 (2022).
2. Muzumdar, M. D., Tasic, B., Miyamichi, K., Li, L. & Luo, L. A global double-fluorescent Cre reporter mouse. *genesis* **45**, 593–605 (2007).
3. Vanlandewijck, M. *et al.* A molecular atlas of cell types and zonation in the brain vasculature. *Nature* **554**, 475–480 (2018).
4. Zhang, Y., Ortsäter, H., Martinez-Corral, I. & Mäkinen, T. Cdh5-lineage-independent origin of dermal lymphatics shown by temporally restricted lineage tracing. *Life Science Alliance* **5**, (2022).
5. Jensen, P. & Dymecki, S. M. Essentials of Recombinase-Based Genetic Fate Mapping in Mice. in *Mouse Molecular Embryology: Methods and Protocols* (ed. Lewandoski, M.) 437–454 (Springer US, Boston, MA, 2014). doi:10.1007/978-1-60327-292-6_26.
6. Vanlandewijck, M. *et al.* A molecular atlas of cell types and zonation in the brain vasculature. *Nature* **554**, 475–480 (2018).

7. Wakabayashi, T. & Naito, H. Cellular heterogeneity and stem cells of vascular endothelial cells in blood vessel formation and homeostasis: Insights from single-cell RNA sequencing. *Front. Cell Dev. Biol.* **11**, (2023).
8. Trimm, E. & Red-Horse, K. Vascular endothelial cell development and diversity. *Nat Rev Cardiol* **20**, 197–210 (2023).
9. Zarkada, G. *et al.* Specialized endothelial tip cells guide neuroretina vascularization and blood-retina-barrier formation. *Dev Cell* **56**, 2237-2251.e6 (2021).
10. Lee, H.-W., Shin, J. H. & Simons, M. Flow goes forward and cells step backward: endothelial migration. *Exp Mol Med* **54**, 711–719 (2022).
11. Lee, H.-W. *et al.* Role of Venous Endothelial Cells in Developmental and Pathologic Angiogenesis. *Circulation* **144**, 1308–1322 (2021).
12. Ekman, N. *et al.* Bmx Tyrosine Kinase Is Specifically Expressed in the Endocardium and the Endothelium of Large Arteries. *Circulation* **96**, 1729–1732 (1997).
13. Tamagnone, L. *et al.* BMX, a novel nonreceptor tyrosine kinase gene of the BTK/ITK/TEC/TXK family located in chromosome Xp22.2. *Oncogene* **9**, 3683–3688 (1994).
14. dela Paz, N. G. & D'Amore, P. A. Arterial versus venous endothelial cells. *Cell Tissue Res* **335**, 5–16 (2009).
15. Bernier-Latmani, J. *et al.* DLL4 promotes continuous adult intestinal lacteal regeneration and dietary fat transport. *J Clin Invest* **125**, 4572–4586 (2015).
16. Bernier-Latmani, J. *et al.* ADAMTS18⁺ villus tip telocytes maintain a polarized VEGFA signaling domain and fenestrations in nutrient-absorbing intestinal blood vessels. *Nat Commun* **13**, 3983 (2022).
17. Bernier-Latmani, J. *et al.* Apelin-driven endothelial cell migration sustains intestinal progenitor cells and tumor growth. *Nat Cardiovasc Res* **1**, 476–490 (2022).
18. Niec, R. E. *et al.* Lymphatics act as a signaling hub to regulate intestinal stem cell activity. *Cell Stem Cell* **29**, 1067-1082.e18 (2022).
19. Wiggins, B. G. *et al.* Endothelial sensing of AHR ligands regulates intestinal homeostasis. *Nature* **621**, 821–829 (2023).
20. Kalucka, J. *et al.* Single-Cell Transcriptome Atlas of Murine Endothelial Cells. *Cell* **180**, 764-779.e20 (2020).
21. Yamamoto, H. *et al.* Integrin β 1 controls VE-cadherin localization and blood vessel stability. *Nat Commun* **6**, 6429 (2015).

22. Su, T. *et al.* Single-cell analysis of early progenitor cells that build coronary arteries. *Nature* **559**, 356–362 (2018).
23. Chavkin, N. W. *et al.* Endothelial cell cycle state determines propensity for arterial-venous fate. *Nat Commun* **13**, 5891 (2022).
24. Fang, J. S. *et al.* Shear-induced Notch-Cx37-p27 axis arrests endothelial cell cycle to enable arterial specification. *Nat Commun* **8**, 2149 (2017).
25. Luo, W. *et al.* Arterialization requires the timely suppression of cell growth. *Nature* **589**, 437–441 (2021).
26. Marcelo, K. L., Goldie, L. C. & Hirschi, K. K. Regulation of endothelial cell differentiation and specification. *Circ Res* **112**, 1272–1287 (2013).
27. Pontes-Quero, S. *et al.* High mitogenic stimulation arrests angiogenesis. *Nat Commun* **10**, 2016 (2019).
28. Rennel, E. *et al.* Endocan is a VEGF-A and PI3K regulated gene with increased expression in human renal cancer. *Exp Cell Res* **313**, 1285–1294 (2007).
29. Shin, J. W., Huggenberger, R. & Detmar, M. Transcriptional profiling of VEGF-A and VEGF-C target genes in lymphatic endothelium reveals endothelial-specific molecule-1 as a novel mediator of lymphangiogenesis. *Blood* **112**, 2318–2326 (2008).
30. Rocha, S. F. *et al.* Esm1 Modulates Endothelial Tip Cell Behavior and Vascular Permeability by Enhancing VEGF Bioavailability. *Circulation Research* **115**, 581–590 (2014).
31. Tammela, T. *et al.* Blocking VEGFR-3 suppresses angiogenic sprouting and vascular network formation. *Nature* **454**, 656–660 (2008).
32. Bahar Halpern, K. *et al.* Lgr5⁺ telocytes are a signaling source at the intestinal villus tip. *Nat Commun* **11**, 1936 (2020).
33. McCarthy, N. *et al.* Distinct Mesenchymal Cell Populations Generate the Essential Intestinal BMP Signaling Gradient. *Cell Stem Cell* **26**, 391-402.e5 (2020).
34. Karlsson, L., Lindahl, P., Heath, J. K. & Betsholtz, C. Abnormal gastrointestinal development in PDGF-A and PDGFR- α deficient mice implicates a novel mesenchymal structure with putative instructive properties in villus morphogenesis. *Development* **127**, 3457–3466 (2000).
35. Huycke, T. R. *et al.* Patterning and folding of intestinal villi by active mesenchymal dewetting. *Cell* **187**, 3072-3089.e20 (2024).

36. Koenig, A. L. *et al.* Vegfa signaling promotes zebrafish intestinal vasculature development through endothelial cell migration from the posterior cardinal vein. *Developmental Biology* **411**, 115–127 (2016).
37. Deng, H., Eichmann, A. & Schwartz, M. A. Fluid Shear Stress–Regulated Vascular Remodeling: Past, Present, and Future. *Arteriosclerosis, Thrombosis, and Vascular Biology* **45**, 882–900 (2025).
38. Pries, A. R. & Secomb, T. W. Making microvascular networks work: angiogenesis, remodeling, and pruning. *Physiology (Bethesda)* **29**, 446–455 (2014).
39. Lizen, B., Claus, M., Jeannotte, L., Rijli, F. M. & Gofflot, F. Perinatal induction of Cre recombination with tamoxifen. *Transgenic Res* **24**, 1065–1077 (2015).
40. Hamilton, T. G., Klinghoffer, R. A., Corrin, P. D. & Soriano, P. Evolutionary divergence of platelet-derived growth factor alpha receptor signaling mechanisms. *Mol Cell Biol* **23**, 4013–4025 (2003).
41. Korsunsky, I. *et al.* Fast, sensitive and accurate integration of single-cell data with Harmony. *Nat Methods* **16**, 1289–1296 (2019).
42. McInnes, L., Healy, J. & Melville, J. UMAP: Uniform Manifold Approximation and Projection for Dimension Reduction. Preprint at <https://doi.org/10.48550/arXiv.1802.03426> (2020).

Figure legends

Reviewer1 Figure1. (a) Endogenous Tdtomato signal (yellow) from *Esm1-CreERT2*, *Bmx-CreERT2* and *Aplnr-CreERT2* transgenic mice in the *Rosa26-mTmG* reporter background after 4-OHT administration at E13.5. GFP (green) and CD31 (red). $n = 6$, $n = 5$ and $n = 5$ respectively. Scale bar, 200 μm . **(b)** Whole-mount of male jejunum showing the distribution of GFP⁺ cells (green) in the mature villus vascular network 2 weeks after 4-OHT administration. VEGFR2 (red), EMCN (white), DAPI (blue) and endogenous Tdtomato signal (yellow). $n = 6$. Scale bar, 50 μm .

Reviewer1 Figure2. (a) UMAP plot of the 3 combined datasets with all identified intestinal and mesenteric cell types with expression levels of *Cdh5*, *Prox1*, *Esm1*, *Pdgfrb* and *Cspg4*. **(b)** High magnification images and single focal planes of whole-mount of E18.5 mesentery and intestine shown in Fig. 1d (after 1 day of lineage tracing). GFP (green) and CD31 (red). Yellow arrowhead marks CD31⁺ GFP⁺ sprouting cell and white arrowheads indicate CD31⁺ GFP⁺ capillaries in the vicinity of the mesenteric artery. $n = 6$. Scale bars, 200, 100 and 20 μm . **(c)** Whole-mount of E18.0 intestines showing decreased GFP⁺ cell contribution to arteries after MAZ51 treatment. GFP (green), CD31 (red), SOX17 (white) and DAPI (blue). Single focal planes are shown from the area outlined in white. White arrowheads mark GFP⁺ CD31⁺ vessels. Vehicle $n = 5$; MAZ51 $n = 8$. Scale bars, 80 and 30 μm . **(d)** At E12.5, mesenteric SOX17⁺ capillaries connected to the CMA are *Aplnr*^{low} (arrowheads). *Aplnr-CreERT2* lineage tracing for 24h. GFP (green), SOX17 (white) and CD31 (red). Cranial mesenteric artery (CMA), mesentery (Mes) and intestine (Int) are indicated. $n = 5$. Scale bars, 100 and 30 μm .

Reviewer1 Figure3. (a) Whole-mount of E14.5 and E18.5 mesentery and intestine after *Esm1* lineage tracing starting at E13.5. GFP (green), CD31 (red) and SOX17 (white). Yellow

arrowheads mark GFP⁺ cells in mesenteric arteries. Mesentery (Mes) and Intestine (Int) are indicated. Scale bar, 200 μm . **(b)** Proportion of GFP⁺ mesenteric arteries after 4-OHT at E13.5 (μm^2 , normalized to vessel area). P value, 2-tailed unpaired Student's *t* test. Error bars, Mean \pm SD. **(c)** Whole-mount of E14.5 and E18.5 mesentery and intestine after Bmx lineage tracing starting at E13.5. GFP (green), CD31 (red) and SOX17 (white). White arrowheads mark CD31⁺ GFP⁻ mesenteric arteries close to the intestine. Mesentery (Mes) and Intestine (Int) are indicated. Scale bar, 200 μm . **(d)** Proportion of GFP⁺ mesenteric arteries after 4-OHT at E13.5 (μm^2 , normalized to vessel area). P value, 2-tailed unpaired Student's *t* test. Error bars, Mean \pm SD. **(e)** Whole-mount of E14.5 and E18.0 mesentery and intestine after Aplnr lineage tracing starting at E13.5. GFP (green), CD31 (red) and SOX17 (white). Yellow arrowheads mark GFP⁺ cells in mesenteric arteries. Mesentery (Mes), Intestine (Int), Vein (Ve) and Artery (Ar) are indicated. Scale bar, 200 μm . **(f)** Proportion of GFP⁺ cells in mesenteric arteries leaving the intestine after 4-OHT at E13.5 (μm^2 , normalized to vessel area). The E18.0 time point result is also shown in Supplementary Fig. 4g. P value, 2-tailed unpaired Student's *t* test. Error bars, Mean \pm SD. **(g)** Intestinal Esm1-derived and Aplnr-derived cells give rise to Bmx⁺ arterial ECs in the submucosal and mesenteric arterial network.

Reviewer1 Figure4. **(a)** Whole-mount of E14.5 mesentery and intestine after Esm1 lineage tracing starting at E13.5. GFP (green), CD31 (red) and CASP3 (white). Yellow arrowheads mark GFP⁺ cells in mesenteric arteries, white arrowhead indicates CASP3⁺ cell. *n* = 2. Scale bars, 100, 30 and 10 μm . **(b)** Quantification of CASP3 intensity (A.U.) normalized to blood vessel (CD31⁺) or GFP⁺ volume (μm^3). **(c)** Whole-mount of E14.0 mesentery and intestine after Esm1 lineage tracing for 10h. GFP (green), ERG (red), Esm1 (white) and Tdtomato (yellow). White arrowheads mark GFP⁻ Esm1⁺ cells, orange arrowhead marks GFP⁺ Esm1⁺ cell and green arrowhead marks GFP⁺ Esm1⁻ cell. Mesentery (Mes) and intestine (Int) are indicated.

$n = 4$. Scale bars, 300, 50 and 25 μm . **(d)** Proportion of ERG^+ ECs labeled for both $\text{GFP}^+/\text{Esm1}^+$, only GFP^+ or only Esm1^+ . P values, 1-way ANOVA with Tukey post-hoc test; Error bars, Mean \pm SD. **(e)** Proportion of GFP^+ cells co-staining with Esm1 and proportion of Esm1^+ cells co-staining with GFP^+ . P value, 2-tailed unpaired Student's t test; Error bars, Mean \pm SD.

Reviewer1 Figure5. (a) UMAP plot of the 3 combined datasets with all identified blood vessel ECs with expression levels of *Nrp2* and *Igfbp3*. **(b)** UMAP plots showing expression levels of venous (*Fam174b*), arterial (*Bmx*) and capillary markers (*Pcdh12*, *Gpihbp1*, *Prdm1*).

Reviewer1 Figure6. (a) Whole-mount of E18.0 intestine 48h after *Esm1-CreERT2*-mediated reporter activation during vehicle or MAZ51 treatment. GFP (green), CD31 (red), CASP3 (white) and DAPI (blue). Arrowheads mark CASP3^+ cells. Vehicle $n = 7$; MAZ51 $n = 4$. Scale bars, 20 and 10 μm . **(b)** Quantification of CASP3 intensity (A.U.) normalized to GFP^+ volume (μm^3). P values, 2-tailed unpaired Student's t test; Error bars, Mean \pm SD.

Reviewer1 Figure7. (a) *Bmx-CreERT2*-controlled GFP expression (green) predominantly marks $\text{CD31}^+ \text{SOX17}^+$ (white) arteries in E14.5 mesenteries. CD31 (red). $n = 4$. Scale bars, 200, 60 and 20 μm .

Reviewer1 Figure8. (a) EMCN (red) is expressed in the venous side of the villus vascular network in the duodenum of adult (12-week-old) males. CAV1 (green), EMCN (red) and DAPI (blue). $n = 5$. Scale bar, 50 μm . **(b)** EMCN (red) is expressed in the embryonic villus capillary network and downregulated in the submucosal arteries. CAV1 (green), SOX17 (white) and

DAPI (blue). $n = 5$. Scale bar, 30 μm . **(c)** UMAP plots of the 3 combined datasets with all identified blood vessel ECs with expression levels of *Bmx*, *Emcn* and *Cav1*.

Reviewer1 Figure9. **(a)** UMAP plot of our embryonic intestinal blood ECs combined with adult intestinal blood EC scRNA-seq datasets from Kalucka et al., Niec et al., and Wiggins et al.^{18–20} **(b)** Heatmaps showing top upregulated genes in embryonic and adult intestinal EC subclusters. **(c)** UMAP plots of the 4 combined datasets showing the expression levels of *Cdh5*, *Esm1*, *Aqp7* and capillary markers *Prdm1*, *Pcdh12* and *Rgcc*. **(d)** UMAP plot of the 4 combined datasets with expression levels of arterial (*Bmx*, *Gja4*, *Hey1*) and venous markers (*Nr2f2* (*Coup-TFII*), *Madcam1*, *Fam174b*). **(e)** Heatmaps showing top upregulated genes in embryonic and adult *Esm1*⁺ EC populations. **(f)** UMAP plots showing the expression levels of *Mest*, *Nid2*, *Adam19*, *Jund*, *Car4* and *Ramp3*.

Reviewer2 Figure1. **(a)** UMAP plot of intestinal blood ECs combined with retinal blood EC scRNA-seq datasets from Zarkada et al.⁹ **(b)** Volcano plot of differentially expressed genes between *Esm1*⁺ ECs from embryonic intestine and P6 wildtype retinal ECs. Genes marked inside green square are not differentially expressed and shared by both retinal and intestinal *Esm1*⁺ ECs. **(c)** Whole-mount staining for GFP (green), CD31 (purple) and DAPI (blue) in P6 *Itgb1*^{iEsm1KO} retinas shows that *Itgb1* deficiency in *Esm1*⁺ cells impairs retinal sprouting angiogenesis. *Itgb1*^{iEsm1HET} $n = 6$, *Itgb1*^{iEsm1KO} $n = 5$. Scale bars, 300 and 50 μm . **(d)** Quantification of P6 retina vascular area (in μm^2 , CD31 area normalized to DAPI area) and GFP⁺ area (in μm^2 , GFP⁺ area normalized to vascular (CD31) or tissue (DAPI) area) in *Itgb1*^{iEsm1HET} and *Itgb1*^{iEsm1KO} mutant littermates. *Itgb1*^{iEsm1HET} group set to 1. P values, 2-tailed unpaired Student's *t* test; Error bars, Mean \pm SD. **(e)** *Itgb1* loss in *Esm1*-expressing cells does not impair intestinal villus development. Whole-mount of E18.5 villus with high magnification

images stained for GFP (green), CD31 (red) and SOX17 (white). *Itgb1*^{iEsm1HET} $n = 3$; *Itgb1*^{iEsm1KO} $n = 3$. Scale bars, 30 and 10 μm .

Reviewer2 Figure2. (a) UMAP plot of our blood ECs combined with retinal blood EC scRNA-seq datasets from Zarkada et al.⁹ (b) UMAP plots with expression levels of *Esm1*, arterial and venous markers (*Bmx*, *Sox17* and *Aplnr*), and proliferation markers (*Mki67* and *Cdk1*). (c) *Itgb1* loss in *Esm1*-expressing cells affects mesenteric artery development, leading to increased arterial cell proliferation. Whole-mount of E17.0 mesenteries for CD31 (green), SOX17 (red) and EdU (white). A mask was applied to identify EdU⁺ SOX17⁺ cells (yellow). Arrowheads mark EdU⁺ SOX17⁺ cells. Control $n = 3$; *Itgb1*^{iEsm1KO} $n = 7$. Scale bar, 30 μm . (d) Graph shows proportion of EdU⁺ SOX17⁺ arterial ECs in control and *Itgb1*^{iEsm1KO} mutant mesenteric arteries. P values, 2-tailed unpaired Student's *t* test; Error bars, Mean \pm SD.

Reviewer2 Figure3. (a) UMAP plot of our 3 combined datasets with all identified intestinal and mesenteric cell types (b) Whole-mount of E18.5 *Pdgfra-eGFP* intestines⁴⁰. GFP (green) and CD31 (blue). $n = 4$. Scale bars, 20 and 5 μm . (c) UMAP plots of intestinal and mesenteric cells with expression levels of intestinal telocyte genes (*Pdgfra*, *Vegfa*, *Bmp2* and *Bmp4*).

Reviewer2 Figure4. Long-term tamoxifen treatment using *Esm1-CreERT2* mouse line generates mosaic mesenteric arteries composed of intestinal *Esm1*-derived ECs (Green cells). Long-term *Itgb1* deletion in *Esm1*-derived cells (*Itgb1*^{iEsm1KO}) leads to cell rounding and reduced mesenteric artery diameter.

Reviewer2 Figure5. (a, b) UMAP plot of our 3 combined datasets with all identified intestinal and mesenteric cell types (a) or blood vessel ECs only (b). (c) VEGFR2 (green) is expressed

in the whole CD31⁺ (red) villus capillary network. Whole-mount of E18.0 intestine. $n = 3$. Scale bar, 20 μm . **(d)** UMAP plots of our 3 combined datasets with only blood ECs with expression levels of *Flt1*, *Kdr*, *Flt4*, *Nrp1* and *Nrp2*. **(e)** UMAP plots of our 3 combined datasets with all identified intestinal and mesenteric cell types with expression levels of *Vegfa*, *Vegfb*, *Vegfc* and *Vegfd*.

Method for the embryonic intestinal scRNA-seq integration with adult intestinal endothelial tip cells.

We obtained scRNA-seq data of several adult small intestine datasets (Niec¹⁸, Kalucka²⁰, Wiggins¹⁹) as FASTQ files. Reads were mapped against the same reference genome and annotations as the data generated in this study using STARsolo (v.2.7.11b) with parameters adjusted for the different technologies used (Kalucka et al.: 10X v2, Niec et al. and Wiggins et al.: 10X v3).

Cells in each dataset were filtered for high mitochondrial content (>10%) and minimum number of counts (Niec, Kalucka: 1000, Wiggins: 2000). All further processing was identical to the data generated in this study (see manuscript Materials and Methods).

Final datasets were merged using AnnData “concatenate” using an outer join. For 2D embedding and clustering, the gene expression matrix was subset to 3000 highly variable genes (HVGs) were selected (sc.pp.highly_variable_genes, flavor “seurat”), and the top 100 principal components (PCs) were calculated. PCs were batch-corrected using Harmony (0.0.5)⁴¹ and served as basis for k-nearest neighbor calculation (sc.pp.neighbors, n_neighbors=30), which were used as input for UMAP⁴² layout (sc.tl.umap, min_dist=0.3) and clustering (scanpy.tl.leiden) at resolution 0.05.

Finally, only endothelial cell (EC) clusters were selected and 2D embedding and clustering were repeated (2000 HVGs, 50PCs, Harmony-correction). The final chosen clustering resolution to delineate EC subtypes was 1.2.

Reviewer1 Figure 1

Reviewer1 Figure 2

a

Reviewer1 Figure 2

Reviewer1 Figure 3

Reviewer1 Figure 4

Reviewer1 Figure 5

Reviewer1 Figure 7

E13.5 → E14.5

Reviewer1 Figure 8

Reviewer2 Figure 1

Reviewer2 Figure 2

Reviewer2 Figure 3

Reviewer2 Figure 4

Long-term
Itgb1 loss in *Esm1*-derived cells

Reviewer2 Figure 5